



# Optimizing High-Resolution Community Earth System Model on a Heterogeneous Many-Core Supercomputing Platform (CESM-HR_sw1.0)

Shaoqing Zhang[1,4,5], Haohuan Fu[*2,3,1], Lixin Wu[*4,5], Yuxuan Li[6], Hong Wang[1,4,5], Yunhui Zeng[7], Xiaohui Duan[3,8], Wubing Wan[3], Li Wang[7], Yuan Zhuang[7], Hongsong Meng[3], Kai Xu[3,8], Ping Xu[3,6], Lin Gan[3,6], Zhao Liu[3,6], Sihai Wu[3], Yuhu Chen[9], Haining Yu[3], Shupeng Shi[3], Lanning Wang[3,10], Shiming Xu[2], Wei Xue[3,6], Weiguo Liu[3,8], Qiang Guo[7], Jie Zhang[7], Guanghui Zhu[7], Yang Tu[7], Jim Edwards[1,11], Allison Baker[1,11], Jianlin Yong[5], Man Yuan[5], Yangyang Yu[5], Qiuying Zhang[1,12], Zedong Liu[9], Mingkui Li[1,4,5], Dongning Jia[9], Guangwen Yang[1,3,6], Zhiqiang Wei[9], Jingshan Pan[7], Ping Chang[1,12], Gokhan Danabasoglu[1,11], Stephen Yeager[1,11], Nan Rosenbloom [1,11], and Ying Guo[7]

[1] International Laboratory for High-Resolution Earth System Model and Prediction (iHESP), Qingdao, China
[2] Ministry of Education Key Lab. for Earth System Modeling, and Department of Earth System Science, Tsinghua University, Beijing, China
[3] National Supercomputing Center in Wuxi, Wuxi, China
[4] Laboratory for Ocean Dynamics and Climate, Qingdao Pilot National Laboratory for Marine Science and Technology, Qingdao, China
[5] Key Laboratory of Physical Oceanography, the College of Oceanic and Atmospheric Sciences & Institute for Advanced Ocean Study, Ocean University of China, Qingdao, China
[6] Department of Computer Science & Technology, Tsinghua University, Beijing, China
[7] Computer Science Center & National Supercomputer Center in Jinan, Jinan, China
[8] School of Software, Shandong University, Jinan, China
[9] Dept of Supercomputing, Qingdao Pilot National Laboratory for Marine Science and Technology, Qingdao, China
[10] College of Global Change and Earth System Science, Beijing Normal University, Beijing, China
[11] National Center for Atmospheric Research, Boulder, Colorado, USA
[12] Department of Oceanography, Texas A&M University, College Station, Texas, USA

*Correspondence to:* Haohuan Fu (haohuan@tsinghua.edu.cn), Lixin Wu (lxwu@ouc.edu.cn)

**Abstract.** With the semi-conductor technology gradually approaching its physical and heat limits, recent supercomputers have adopted major architectural changes to continue increasing the performance through more power-efficient heterogeneous many-core systems. Examples include Sunway TaihuLight that has four Management Processing Element (MPE) and 256 Computing Processing Element (CPE) inside one processor and Summit that has two central processing units (CPUs) and 6 graphics processing units (GPUs) inside one node. Meanwhile, current high-resolution Earth system models that desperately require more computing power, generally consist of millions of lines of legacy codes developed for traditional homogeneous multi-core processors and cannot automatically benefit from the advancement of supercomputer hardware. As a result, refactoring and optimizing the legacy models for new architectures become a key challenge along the road of taking advantage of greener and faster supercomputers, providing better support for the global climate research community and contributing to the long-lasting society task of addressing long-term climate change. This article reports the efforts of a



large group in the International Laboratory for High-Resolution Earth System Prediction (iHESP) established by the cooperation of Qingdao Pilot National Laboratory for Marine Science and Technology (QNLM), Texas A & M University and the National Center for Atmospheric Research (NCAR), with the goal of enabling highly efficient simulations of the high-resolution (25-km atmosphere and 10-km ocean) Community Earth System Model (CESM-HR) on Sunway TaihuLight.

The refactoring and optimizing efforts have improved the simulation speed of CESM-HR from 1 SYPD (simulation years per day) to 3.4 SYPD (with output disabled), and supported several hundred years of pre-industrial control simulations. With further strategies on deeper refactoring and optimizing for a few remaining computing hot spots, we expect an equivalent or even better efficiency than homogeneous CPU platforms. The refactoring and optimizing processes detailed in this paper on the Sunway system

should have implications to similar efforts on other heterogeneous many-core systems such as GPU-based high-performance computing (HPC) systems.

## 1 Introduction

The development of numerical simulations and the development of modern supercomputers have been like two entangled streams, with numerous interactions at different stages. The very first electronic

computer, ENIAC (Lynch, 2008), produced the simulation of the first numerical atmospheric model in the 1950s, with a spatial resolution of 500-800 km. In the 1960s, the first supercomputer (as compared to regular electronic computer), CDC 3300 and 6600, designed by Seymour Cray, was installed at the National Center for Atmosphere Research (NCAR) to provide large-scale computing to the U. S. national community engaging in atmospheric and related research. Since then, a constant increase in

both the computing power of supercomputers and complexity of numerical models has been witnessed by the global community. Over the decades, the peak performance of a supercomputer system has already evolved from the scale of 1 Mflops (e.g., CDC 3300) to 100 Pflops (e.g., Sunway TaihuLight), which is an increase of 11 orders of magnitude. Meanwhile, the numerical weather model has also evolved into a highly-complex Earth system model, with increased resolution and better representation

of physics and interpretation of all the scientific knowledge accumulated over the decades.
One major development of the supercomputing hardware in recent years is the architectural changes of the processors. Due to the physics limitations, the regular increase of frequency came to a stop roughly one decade ago. Since then, the increase of computing power is largely from an increased density of computing units in the processors. As a result, for the leading-edge supercomputers that were developed

after 2010, a major part of computing power is provided by many-core accelerators, such as NVIDIA GPUs (Vazhkudai et al., 2018), Intel Xeon Phi MICs (Liao et al., 2014), and even reconfigurable FPGAs (Gan et al., 2017; Chen, 2019). As a result, recent machines contain two major architectural changes. One change is from a homogeneous architecture to a heterogeneous architecture, for which one now faces an environment with multiple types of computing devices and cores, instead of

programming and scheduling a single kind of core in a system. The other change is the significant increase in the number of cores, which usually comes with a reduced complexity of each single core. The huge number of available parallel cores requires a complete rethinking of algorithms and data structures, which is a huge challenge when taking advantage of the new supercomputers.





Climate science advances and societal needs require Earth system modelling to resolve more details of
geo-physical processes in the atmosphere, ocean, sea-ice and land-soil components. However, the
model resolution must be sufficiently fine to better resolve regional changes/variations as well as
extreme events (Delworth et al., 2012; Small et al., 2014; Roberts et al., 2018). Scientifically, there are
two important questions that need further understanding in current Earth climate sciences: 1) how
global and large-scale changes/variations influence the local weather-climate anomalies, and 2) how
local weather-climate perturbations feedback to large-scale background. To address these questions,
Earth system models must be able to explicitly resolve more and more local fine-scale physical
processes with higher and higher resolutions. While such high-resolution modelling efficiently advances
the understanding of the attribution and impact of large-scales such as global changes, it also provides
an opportunity to link scientific advances with local severe weather-climate alerts, and promote societal
services.  Given that the model resolution is intractable with computing resources available, higher and
higher resolution Earth modelling demands greener supercomputing platforms with more affordable
energy consumption.
In the recent decade, with the power consumption of a single supercomputer surpassing the boundary of
10 MW (equivalent to the power consumption of three world-class universities like Tsinghua), the
energy issue has become a major factor in both the design and the operation process. As a result,
discussions are raised for the community of climate change studies. Taking CMIP (e.g. Meehl et al.,
2000) as an example, while the goal is to reduce emission and to mitigate global warming effects, the
huge computing cost and the related power consumption are also generating a considerably large carbon
footprint. Similar concerns are also expressed for recent artificial intelligence (AI) efforts that utilize
enormous computing resources to train various deep neural networks (e.g. Schwartz et al., 2019), and
ideas are proposed to report the financial cost or "price tag" for performing such research operations.
Therefore, reducing the energy consumption is the other major factor that drives us to port and to
refactor the model to new architectures. Comparing many-core accelerators with traditional multi-core
processors, while the computing power generally increases by 10 to 20 times, the power consumption
only expands by several times, leading to a significantly more efficient power solution.
While we have stated multiple benefits for redesigning the models for new supercomputer platforms,
the challenge, especially in the coding parts, is also many-fold. The first major challenge comes from
the heavy legacy of the climate code (Fu et al., 2017. With the evolution of the Earth system as one of
the most challenging and complicated scientific problem, it involves generations of scientists from all
over world to resolve. Most of the scientists only focus on a sub-problem and contribute to a subroutine
of the entire model. As a result, the current Earth system model consists of hundreds of modules that
were written at different times and even with different versions of programming languages. The
complexity in both the code and the science is a challenge for moving the code to different and more
modern architectures.
The second major challenge comes from the computation load pattern of the code. As the model
involves a large number of kernels that describe different parts of the climate system, when we look at
the execution pattern, there are no "hot spots" that consume a large portion of the computational cycles.
Instead, we have hundreds of hot spots, each of which consumes a small portion of the computation
cycles. Therefore, to achieve performance benefits through the porting process, we need to take all of
these hundreds of modules into considerations. In contrast, for some programs with large hot spots such





as an earthquake simulation code (You et al., 2013), we only need to study several kernels that consume over 90% of the computing cycles to achieve notable performance benefits.

The third challenge comes from the dramatic architectural change. As we mentioned above, the transition from multi-core CPUs to many-core processors or accelerators brings completely different requirements for both parallelism and data locality. Therefore, the entire process not only involves simple porting of corresponding instructions and statements, but also requires a complete rethinking from the algorithm level to achieve both suitable parallelism and suitable caching of variables for the new underlying hardware architecture.

With the GPU devices introduced as accelerators for general-purpose computing around 2007 (Owens et al., 2007), most early efforts related to climate or weather modelling only focused on certain parts of the models, mostly physics or chemical parameterization schemes that are structurally suitable for parallel architectures, such as the chemical kinetics modules (Linford et al., 2009) and the microphysics scheme (Mielikainen et al., 2013) in WRF (Weather Research and Forecasting model), the shortwave radiation parameterization of CAM5 (the 5[th] version of Community Atmospheric Model) (Kelly, 2010), and the microphysics module of Global/Regional Assimilation and Prediction System (GRAPES) (Xiao et al., 2013). With both a high parallelism and a high arithmetic density, these modules demonstrate a high speedup (ranging from 10 to 70 or even to 140x) when migrating from CPU to GPU. In contrast, the dynamical core code, which involves both time-stepping integration and communication among different modules is more challenging to port to heterogeneous accelerators. Examples include NIM (Govett et al., 2010), GRAPES (Wang et al., 2011), CAM-SE (Carpenter et al., 2013), HIRLAM (Vu et al., 2013), and NICAM (Demeshko et al., 2012), with a speedup of 3 to 10x. For migrating an entire model, the efforts are even fewer to see. One early example is an GPU-based acceleration of ASUCA, which is the next-generation high resolution mesoscale atmospheric model developed by the Japan Meteorological Agency (JMA) (Shimokawabe et al., 2010), with a speedup of 80-fold with a GPU compared to a single CPU core. In recent years, we see complete porting of several models from CPU to the GPU platforms, such as the GPU-based Princeton Ocean Model (POM) (Xu et al., 2014) and the GPU-based COSMO regional weather model by MeteoSwiss (Fuhrer et al., 2018). For a porting of a model at such level, these three challenges mentioned above (heavy code legacy, hundreds of hot spots distributed through the code, and the mismatch between the existing code and the emerging hardware), have apparently combined to make more challenges. Facing the problem of tens of thousands of lines of code, the researchers and developers have to either perform an extensive rewriting of the code (Xu et al., 2014) or invest years of effort into redesign methodologies and tools (Gysi et al., 2015).

This article reports the efforts of a large group in International Laboratory for High-Resolution Earth System Prediction (iHESP) established by the cooperation of Qingdao Pilot National Laboratory for Marine Science and Technology (QNLM), Texas A & M University, and the National Center for Atmospheric Research (NCAR), with the goal of enabling highly efficient simulation of the Community Earth System Model high-resolution version (25-km atmosphere and 10-km ocean) (CESM-HR) on Sunway TaihuLight. As far as we know, this may be the first time to tackle a complete global high-resolution climate system model, instead of a regional model that usually focuses on short-term events. With the early-stage efforts focusing on the atmospheric component CAM5 (Fu, et al., 2016a; Fu et al., 2017), the work in this article, finally extends to the complete scope of CESM-HR and creates a redesigned software with millions lines of codes transformed to a heterogeneous many-core





architecture, and to deliver substantial performance benefits. Compared with the existing work mentioned above (GPU-based ASUCA, POM, WRF, and COSMO), our work optimizes an Earth
system model, which demonstrates a next-level of complexity in both components and numerical methods, and requires better accuracy and better conservation of matter and energy, so as to perform simulation of hundreds of years instead of just hundreds of days. The refactoring and optimizing efforts have improved the simulation speed of CESM-HR from 1 SYPD (simulation year per day) to 3.4 SYPD, and supported several years of pre-industrial control simulations.
The rest of the paper is organized as follows. After the introduction, Section 2 describes the major features of the Sunway TaihuLight Supercomputer, including its architecture and energy consumption details, and the corresponding parallelization strategies etc. Section 3 gives detailed methods of refactoring and optimizing CESM-HR on Sunway TaihuLight. Section 4 demonstrates stable and sound scientific results of the first few hundred years of CESM-HR simulations on the new architecture.
Finally, the summary and discussions are given in Section 5.

## 2 A heterogenous many-core supercomputer: Sunway TaihuLight

### 2.1 The design philosophy of the Sunway machine

#### 2.1.1 Hardware

**1) The new processor architecture.** Considered a major milestone in the HPC development history of
China, Sunway TaihuLight (Fu, et al, 2016b; Dongarra, 2016) is the Chinese first top one system on the Top500 (https://www.top500.org/lists/2019/11/) list that is built using Chinese homegrown processors. The heterogeneous many-core processor, SW26010 (as shown in the name, a 260-core CPU), provides all the computing capabilities of the system. Each SW26010 processor, as shown in Fig. 1, can be divided into 4 identical core groups (CGs), which are connected through the network on chip. Each CG
includes one management processing element (MPE), one computing processing element (CPE) cluster with 8x8 CPEs and one memory controller that shares and manages the memory bandwidth to 8GB DDR3 memory. Within the 8×8 mesh, the CPEs can transfer data along the rows or columns through the low-latency register communication features, which provide low-cost data sharing schemes within the cluster. The running frequency of each element is 1.45 GHz.
**2) A different memory hierarchy.** Most current scientific computing models are constrained by the memory bandwidth rather than the computing speed. Climate models are typical examples of models that achieve less than 5% of the peak performance of current computers. As a result, a large part of our effort tries to achieve a suitable mapping of the CESM model to the unique memory hierarchy of Sunway TaihuLight. The outermost memory layer is the 32GB DDR3 memory equipped in each
compute node, shared among the four CGs and 260 cores, which is quite similar to the DDR memory in traditional IBM or Intel CPU machines. A major difference is that the same level of DDR3 memory bandwidth needs to be shared among a significantly larger number of cores (from dozens to hundreds). Inside the processor, the MPE, which is designed to provide similar functions to traditional CPUs, also adopts a similar cache hierarchy, with a 32KB L1 instruction cache, a 32KB L1 data cache, and a
256KB L2 cache for both instruction and data. The major changes are in the CPEs. To meet the



hardware and power constraints with a maximized level of computing density, inside each CPE, the cache is replaced by a 64KB user-controlled scratchpad memory, called LDM (local data memory). Such a change in cache hierarchy requires a complete rethinking of the data structure and loop structure. Previous programmers could rely on the cache hierarchy to achieve a reasonable buffering of temporary
variables when using OpenMP to start independent threads in different CPU cores. Migrating to Sunway, the programmers need to handle the memory part more elegantly to achieve any meaningful utilization of the system. The scratchpad fast buffer also becomes the last weapon for the programmers to address the proportionally reduced memory bandwidth of the system. In many cases, the manually designed buffering scheme would improve data reuse and increase the computing performance. As a
result, instead of directly reading from the DDR memory, most kernels would load the data into LDM manually, and start the computation from there.

**3) Customized system integration.** Using the SW26010 CPU as the most basic building block, the Sunway TaihuLight system is built by a fully-customized integration at different levels: (1) a computing node with one processor and 32GB memory; (2) a supernode with 256 computing nodes that are tightly
coupled using a fully-connected crossing switch; (3) a cabinet with 4 supernodes; (4) the entire computing system with 40 cabinets. Such an integration approach can provide high performance computing power in a high-density form. With 40 cabinets, the entire Sunway TaihuLight system provides in total $40 \times 4 \times 256 = 40,960$ CPUs, and $40,960 \times 260 = 10,649,600$ parallel cores.

### 2.1.2 Software

**1) The original compiling tools.** Targeting a completely new many-core processor and system with over 10 million cores, the compiling tools are probably the most important tools to support the development of applications on Sunway TaihuLight. The set of compilation tools includes the basic compiler components, such as the C/C++, and Fortran compilers. In addition to that, there is also a parallel compilation tool that supports the OpenACC 2.0 syntax and targets the CPE clusters. The
customized Sunway OpenACC tool supports management of parallel tasks, extraction of heterogeneous code, and description of data transfers. Moreover, according to the specific features of the Sunway processor architecture, the Sunway OpenACC tool has also made several syntax extensions from the original OpenACC 2.0 standard, such as a fine control over buffering of multi-dimensional array, and packing of distributed variables for data transfer. In addition to Sunway OpenACC, the Sunway
platform also provides an Athread interface for the programmers to write specific instructions for both the computing and the memory parts. As discussed in the later sections, different approaches are taken for the redesign of different parts of the CESM model.

**2) The compiling and profiling tools developed along this project.** Earth system models are some of the most complex numerical models that scientists have ever built. Unfortunately, these models are also
some of the very first software that were ported to the new Sunway TaihuLight system, and tested through the relatively new compiler system. As a result, to facilitate the development, a number of new compiling and profiling tools were developed along the way of porting and redesigning the model, with detailed information listed in **Table 1**. For example, the hot spots analysis tool called *swlu* is particularly useful as the Community Earth System Model high-resolution version (CESM-HR) is
refactored and optimized on the Sunway machine. Once the points of *swlu_prof_start* and





*swlu_prof_stop* are set, an output file that contains the detailed information of computing consumption analysis of the targeted code block is produced at the end of the test experiment. This will be discussed in more details Section 3.3 when the CESM-HR is refactored and optimized on Sunway machine.

### 2.1.3 The 2nd level parallelism between MPE and CPEs.

As shown in **Fig. 2**, after the domain-based task decomposition in the traditional Message-Passing-Interface (MPI) parallelism (we refer to this as the 1st-level parallelism) among the core groups, within each core-group, the Sunway machine requires a CPE-based task decomposition and communication between the MPE and CPEs (we refer to this as the 2nd-level parallelism). Since each CPE only has a 64KB scratchpad memory as the fast buffer, each piece of work in the CPE-based task decomposition
needs to be a suitable size for this fast buffer. This is the major challenge to enhance computational efficiency of a model on the Sunway machine, which will be discussed in more details in section 3.3.

### 2.2 Major characteristics

### 2.2.1 Architectural pros and cons

In order to achieve extreme performance and power efficiency, the computing chip of Sunway-
TaihuLight (SW26010 CPU) abandons the cache structure, so as to spare the on-chip resources for more computing occupancy. As a result, each CG of the SW26010 is able to deploy 64 slave cores, and the 64 slave cores can communicate with each other based on register communications. Therefore, a more fine-grained communication mechanism is provided to allow more sophisticated operations and optimizations on dealing with data.
The on-chip heterogeneity of the SW26010 processor enables a uniform memory space between MPE and CPEs to facilitate the data transfer. Such a behaviour is different from the conventional cases such as the CPU-GPU scenario where data transfer has to go between different processors and accelerators. On the other hand, the on-chip heterogeneity also leads to the uniform programming model between MPE and CPEs, and is promising for achieving better overlapping strategies. In that sense, the Sunway
architecture is more plausible for scientific computation like Earth system modelling than a GPU system.

### 2.2.2 Power efficiency

**Table 2** lists the power efficiency of the Sunway TaihuLight system and some major systems around the world containing with different processors such as CPU, GPU, and Xeon Phi, etc. It can be figured
out that the Sunway TaihuLight, due to the lower chip frequency (1.45GHz per SW26010), as well as a sophisticated water cooling system for the whole machine, is greener than some major systems that are widely used for doing climate simulations.



## 3 Enabling CESM-HR on Sunway TaihuLight

### 3.1 An Overview of the Community Earth System Model high-resolution version (CESM-HR)

The CESM version used in the present study is the CESM1.3-beta17_sehires38 tag, representing an extension of the CESM1.3-beta17_sehires20 tag described in Meehl et al. (2019). This latter model tag was developed specifically for supporting a high-resolution CESM version with 0.25° CAM5 (the 5th version of Community Atmospheric Model) atmosphere and 0.1° POP2 (the 2nd version of Parallel Ocean Program) ocean model. The details of this progression from CESM1.1 to CESM1.3-beta17 are

provided in Meehl et al. (2019) where it is concluded that the most impactful developments were in the atmospheric model with changes to vertical advection, the gravity wave scheme, dust parameterization, and move from the Finite Volume (FV) dynamical core to the Spectral Element (SE) dynamical core with better scaling properties at high processor counts. The new atmospheric physics result in better positioning of the Southern Hemisphere jet and improved high and low cloud simulations, with general

increases in low cloud in CESM1.3, producing better agreement with available observations. Notable developments in the CESM1.3 progression from beta17_sehires20 to the current beta17_sehires38 version include two modifications for the POP2 ocean model. The first change involved back-porting a new iterative solver for the barotropic mode from CESM1.2 to CESM1.3 to reduce communication costs, particularly beneficial for high-resolution simulations on large processor counts (e.g., Hu et al.,

2015; Huang at al., 2016). The second modification was a change of the ocean coupling frequency from one hour to 30 minutes to alleviate any potential coupling instabilities that may arise with longer coupling frequencies between the ocean and sea-ice models. In addition, we switched from an older shortwave calculation method to the newer delta-Eddington shortwave computation of Briegleb and Light (2007). Although this latter method had been the default shortwave computation for the standard

resolution (1°) simulations since CCSM4/CESM1, it had not been used in high-resolution simulations. There were also a few sea-ice namelist parameter changes as in Small et al. (2014) that affect the computation of shortwave radiation in the sea ice model for this high-resolution configuration. Finally, because the initial sea-ice thickness and extent were deemed too thin and not extensive enough for a pre-industrial configuration, further adjustments were done to the melting snow grain radius and the

melt onset temperature to increase the sea-ice albedo with the effect of thickening the sea ice and increasing the extent. Note that although these CESM 1.3 tags are informal, they are available to the community upon request via the CESM web pages. The Sunway version of the model is available via GitHub at https://github.com/ihesp/CESM_SW. The structure of the CESM-HR model tells the challenge of the project: migrating a combination of several complex models onto a completely new

architecture. The atmosphere, the ocean, the land, the sea-ice, and the coupler, each of these models can be fairly complicated with hundreds of thousands of lines of code.
Using the analysis tool *swlu* mentioned in section 2.1.2 to produce output file *swlu_prof.dot* and conducting a command such as "dot -Tpdf swlu_prof.dot -o B1850CN_ne120_t12_sw8000.pdf", we obtain the analysis flow-chart as **Fig. 3**. **Figure 3** first shows that the main computing consumption is

for the main body RUN_LOOP in the five parts of the model driver, about 93.53%, and the next is the model initialization, 4.74%. Extending RUN_LOOP, we can see that the major computing consumption





is for the atmosphere ATM_RUN (52.91%) and ocean OCN_RUN (28.73%), and the next is for the sea-ice ICE_RUN (6.96%).

In the entire model, the atmosphere (CAM5) and the ocean (POP2) are still the dominant consumers of computing cycles. Therefore, in the process of migrating towards a new machine, CAM5 and POP2 are the main targets that require refactoring and redesign, with the other components only requiring porting efforts. However, even only considering CAM5 and POP2, the complexity is at a level that requires tremendous efforts. **Figures 4** and **5** show the major hot spots (the functions that consume the most run 320 time, i.e. conducting significant computation) of these two models respectively. Looking into the major functions in both CAM5 and POP2, it is clear that most major computing kernels only consume around 5% of the total run time. In CAM5, the only substantial component that takes more than 10% run time is the *bndry_exchangev* function, which performs the MPI communication between different MPI processes, and takes a substantial part of the total run time when the parallel scale increases to over 20,000 MPI processes. In POP2, the only clear hot spot is the *kpp* function, which performs the 325 computation of K-Profile vertical mixing parameterization (KPP) (Large et al., 1994). Note that, when going further down, these functions would be further decomposed into even more functions at a lower level. As a result, for both CAM5 and POP2, over hundreds of functions were redesigned to achieve a reasonable performance improvement of the entire CESM-HR model

**3.2 Porting of CESM-HR**

**3.2.1 Migrating the code from Intel to Sunway processors**

The new CESM-HR is first run on the Intel multiple-core supercomputing platforms (Intel Xeon E5-2667 CPU) at TAMU and QNLM. The run on Intel CPUs is used as a stable climate simulation with a good radiative balance at the top of atmosphere (TOA). Roughly, using 11,800 Intel CPU cores, the model can achieve a simulation speed of 1.5 SYPD with standard output frequency (monthly or daily 335 3D fields and 6-hr 2D fields). As we know, when a climate model with over a million lines of code such as CESM-HR is ported to a different supercomputing system, due to the differences in computing environment (compiler version, for example), numerical solutions cannot be guaranteed bitwise identical. Therefore, we use the CESM ensemble consistency test (ECT) (e.g., Baker et al., 2015; Milroy et al., 2018) to evaluate simulations run on the TAMU and QNLM Intel multi-core 340 supercomputing platforms. In particular, we use the UF-CAM-ECT test (Milroy et al, 2018) tool from CESM-ECT to determine whether three short test runs (of nine timesteps in length) on the TAMU and QLNM machines are statistically indistinguishable from a large "control" ensemble (1000 members) of simulations generated on NCAR's Cheyenne machine. The general ECT idea is that the large ensemble control ensemble, whose spread is created by round-off-level perturbations to the initial temperature, 345 represents the natural variability in the climate model system and thus can serve as a baseline against which modified simulations can be statistically compared using principal component (PC) analysis. The code ports to the TAMU and QNLM Intel multi-core supercomputing platforms both passed the ECT tests (meaning that the simulations were statistically indistinguishable from the control).

The porting of the CESM-HR from the Intel homogeneous multi-core platform to the Sunway MPE-350 only system began on December 3rd, 2017. The major task of this stage was to resolve compatibility issues of the Sunway and standard Fortran compilers. With support from both the Sunway compiler





team and the maintenance team of the Wuxi National Supercomputing Center, the team spent roughly six months going through over one million lines of CESM code. Starting on October 1st, 2018 on, the Sunway MPE-only version of CESM-HR was able to stably run on Sunway TaihuLight. The timing
results are shown in **Table 3**. Without the utilization of CPEs, the speed is roughly 1 SYPD.

### 3.2.2 Correctness verification

Given that different supercomputing platforms with different architectures generally produce nonidentical arithmetic results (due to the differences from architectural, compiler, rounding and truncation etc.), it is challenging to ensure the correctness of the code porting process between different
machines with different hardware architectures. We initially evaluate the Sunway MPE-only port by looking at the effect of a small $O(10^{-15})$ perturbation in the initial sea surface temperature (SST). **Figs. 6ab** gives the systematic "error" growth in the SST that resulted from such a tiny initial error (mainly over the North Pacific and North Atlantic as well as Southern Ocean and North Indian Ocean) after a few months of model integration due to the model internal variability. **Fig. 6c** shows that the difference
produced by the Sunway MPE-only and the homogeneous multi-core Intel machines are within a similar range of differences produced by perturbation runs (either a round-off initial error or different Intel machines). Then, we again use CESM-ECT to evaluate the Sunway MPE-only version of CESM-HR, using three different compiling optimization options (-O1, -O2 and -O3) and show the results in **Table 4**. Note that by default, CESM-ECT evaluates three simulations for each test scenario and issues
an overall fail (meaning the results are statistically distinguishable) if more than two of the PC scores are problematic in at least two of the test runs. We see that with -O1 and -O3 compiling optimization options, the Sunway MPE-only version passes the ECT test, but it fails with -O2 optimization. This -O2 result is not concerning as the failure with three PCs (out of 50 total) is just outside the passing range and small uncertainties are not uncommon (Milroy et al., 2016).

### 3.3 Refactoring and redesign of CAM5 and POP2: major strategies

### 3.3.1 Transformation of independent loops

In a scientific simulation program, loops are generally the most computing-intensive parts. Especially for the time integration in climate models, a major part is to advance the time step in both dynamic and physics processes. In such processes, the major form is usually independent loops that iterate through all
the different points in a 3D mesh grid. As the computation of each point is generally independent (or in some case dependent on a small region of neighbouring points), assigning different partitions of the loop to different MPI processes or threads is the most important approach of achieving parallelism. Migrating from Intel CPU to Sunway CPU, the approach to parallelism is still the same, with different processes at the first level and different threads at the second level. The major change comes from the
significantly increased parallelism inside each processor, and the architectural change of the cache part. In Intel CPU machines, we have dozens of complex cores, each of which has its own cache and support for a complex set of instructions. In the SW26010 many-core processor, we have 256 CPEs, each of which only has 64KB LDM and a reduced set of computational instructions. Therefore, the challenge





becomes how to remap the loops to fit the increased number of cores and limited scratchpad fast buffer
allocated to each core.

**Figure 7** demonstrates some of the typical loop transformations that we perform for both the dynamic
parts and physics parts, so as to produce the most suitable loop bodies that fit the number of parallel
cores and the size of the fast buffer in each core of the SW26010 many-core processor. The first kind of
transform is a pre-processing step. We aggregate all the loop bodies into the most inner loop, so that
afterwards we can more easily interchange or partition different loops. A typical example is the
*euler_step* loop in the dynamic core of CAM. We perform such an aggregation at the very beginning, so
that all the computation instructions reside in the most inner loop, and the dependencies are reduced to a
minimum level. The second kind of transform, loop interchange, is one of the most frequently used
transforms in our refactoring and redesign of CAM5 and POP. The loop interchange transform is often
used to expose the maximum level of parallelism for the 256 parallel CPEs in each Sunway processor.
The third kind of transform merges two loops into one. The purpose of such a transform is similar to the
second one. If none of the loops can provide enough parallelism for the many-core processor, we
choose to merge some of the loops to release enough parallel threads.

After transforming the loops, in most scenarios, we can achieve a suitable level of both parallelism and
variable storage space for the CPE cluster architecture. Afterwards, we can apply the OpenACC
directives to achieve automated multi-threading of the loops and to allocate different iterations to
different CPEs. However, in certain cases, either to remove certain constraints in the existing algorithms
or to achieve better vectorization, we need to perform a more fine-grained Athread-based approach,
detailed in the following section.

**3.3.2 Register communication based parallelization of dependent loops**

Besides independent loops that process different points in the mesh grid, scientific codes also include
dependent loops that are not straightforward to parallelize on different cores. A typical case is the
summation process. For example, in the *compute_and_apply_rhs* kernel, we need to compute the
geopotential height using the following loop:
for ($i$=1; $i$ <127; $i$++)  $p_i = p_{i-1} + a_i$;
where the initial value $p_0$  (the value of $p_{i-1}$ when $i$=1) is the initial geopotential height, and $a_i$  is the
pressure differences between two neighboring layers $i$ and $i-1$. Such a loop is clearly data dependent, as
the computation of $p_i$ relies on the computation of $p_{i-1}$. Therefore, we can not directly allocate different
iterations to different cores, otherwise each core would wait for the result of the previous element and
cannot compute in parallel.

To achieve parallel computation in such a scenario, we compute the accumulative pressure difference
for the corresponding layers within each CPE, and achieve efficient data transfer among different CPEs
through register communication.  As shown in **Fig. 8**, we compute the result in three stages. Assume
that we parallelize the computation of 128 layers in 8 parallel CPEs, and each CPE would store 16 $p_i$
and $a_i$  values.

**Stage 1, Local Accumulation:** Each CPE computes the local accumulation sum of its own layers.





**Stage2, Partial Sum Exchange:** Each CPE $C_i$ which is not in the first waits for the $p_{16 \times i-1}$ sent from $C_{i-1}$ through the register communication feature, and then computes $p_{16 \times (i+1)-1} = p_{16 \times i-1} + \sum_{j=0}^{15} a_{i+j}$. Afterwards, CPE $C_i$ sends $p_{16 \times (i+1)-1}$ to the next CPE $C_{i+1}$, if $C_i$ is not in the last one in the row.

**Stage 3, Global Accumulation:** Computes all the atmospheric pressures within each CPE.

In such a way, we can maximize the parallel computation happened in each CPE, and utilize the register communication feature to achieve fast data transfer among the different CPEs.

### 3.3.3 Athread-based redesign of the code

Even after identifying the most suitable form of loops to be parallelized, the *openacc* approach
sometimes still provides disappointing performance. For example, in certain kernels in the *euler_step* function, the 64 CPEs would only provide equivalent performance to 1.5 Intel cores. There are multiple reasons behind this poor performance improvement. The most essential factor is that the algorithm and the data structure were all designed for the previous multi-core CPU architecture, and are inherently not suitable for the emerging many-core architecture. As a result, the OpenACC approach, which minimizes
the code changes but also removes the possibility of an algorithmic redesign, leads to a program that would underperform on the new architecture. Therefore, for the cases that OpenACC performs poorly, we take a more aggressive fine-grained redesign approach and rewrite CAM5 using the Athread interface – an explicit programming tool for CPE parallelization. In such an approach, we make careful algorithmic adjustments.
Taking the OpenACC-refactored version as a starting point, we first rewrite the OpenACC Fortran code to the Athread C version. The second step is to perform manual vectorization for certain code regions. We can specifically declare vectorized data types and write vectorized arithmetic operations, so as to improve the vectorization efficiency of the corresponding code regions.

### 3.3.4 Other tuning techniques at a glimpse

*(a). Data reversing for vectorization.* SIMD vectorization (Eichenberger et al., 2004) is an important approach to further explore the parallelism within each core. To achieve a better vectorization performance, we need a careful modification of the code, to achieve rearrangement of either the execution order or the data layout. In certain cases, reversing the order of the data is necessary to have a more optimal pattern.

*(b). Optimizing of DMA (Direct Memory Access).* Directly calling the DMA functions will add additional overhead and lower the overall performance. So in some occasions when the DMA function call becomes bottlenecks, we can try bypassing it using DMA intrinsic for instance.

*(c). Tiling of data.* For certain operations such as stencil computation (Gan et al., 2014) that has complicated data accessing patterns, the computing and accessing can be done in a tiling pattern
(Bandishti et al., 2012) so that the computation of different lines, planes, or cubes, can be piplined and overlapped.

*(d). Fast and LDMsaved math library.* The original math library (e.g., xmath (Zhang et al., 2016)) for the Sunway system is not specifically designed for LDM saving, as data in the LDM may not be released in time after the library is called. So we modified the library to use instant LDM space creation
or releasing, saving more LDM for other usages.





*(e). Taking control logics out of loops.* This is a common and popular idea to reduce the overhead/number of branches, and thus to increase the performance.

*(f). Vertical layer partition.* This is a specific idea for some kernels in the physics part with too many parameters or variables. We can vertically decompose these kernels into several sub-kernels, which are then assigned to several CPEs that are also divided into several groups.

*(g). Grouping of CPEs.* Instead of using all 64 CPEs to deal with one thing, we can separate them into several groups for different simultaneous tasks. (The vertical layer partition mentioned above is an example of using this method.)

*(h). Double buffer.* The DMA double buffer mechanism can help load data ahead of time before it is required for computing. Such a strategy can therefore push data closer to the computation.

*(i). Function inline.* For some functions that have too many calling overheads, we can try to make them inline so that the function calling overhead can be reduced.

### 3.4 Refactoring and optimizing of the CAM5 computing hot spots

Following the profiling procedure described in section 3.1, we identify the bottom modules/subroutines/functions which have significant computing consumption in the CAM5 dynamic core and physics modules. We list the examples of CAM5 physics in **Table 5** by the order of optimization rate shown in the last column. We can see that for most physics modules that are implemented as column-wise computations with no dependent data communications, our refactor and optimization efforts demonstrate a reasonable performance improvement, ranging from 3 to 9x. **Table 5** also demonstrates the major strategies and tuning techniques (Section 3.3.4) used for each hot spot function.

### 3.5 Refactoring and optimizing of the POP2 computing hot spots

Extending the procedure of tracking the computing hot spots into OCN_RUN, we locate the modules that have the most computing consumption as listed in **Table 6**. The last column lists the optimization rate with the CPE-parallelism which is defined as the ratio of the MPE-computing time over CPE-parallelized computing time using 18,300 CGs. The optimization rate of the kpp vertical mixing module (vmix_kpp.F90), which is one of the most time-consuming module in POP, is up to 7.49, demonstrating a well improved efficiency of the refactored and redesigned code. **Table 6** also describes the tuning techniques (Section 3.3.4) applied to each hot spot functions.

### 3.6 Evaluation of numerical differences after applying CPE parallelization of the code

Refactoring and optimizing the majority of CESM-HR resulted in a version that we refer to as SW-CPE CESM-HR.

As the arithmetic behaviours of MPE and CPE in the Sunway processor are not completely identical, depending on nonlinearity levels of computation, the CPE-parallelized code may produce slightly different results from the MPE-only code, with deviation on last a few digits. **Table 7** shows a few examples of accuracy verification of subroutines in major modules of POP2. We can



see that for simple array addition and multiplication (in the barotropic solver module, for instance), the result of the CPE-parallelized code is identical to the result produced by the MPE-only code. However, for multi-level array multiplication or strongly-nonlinear function computation (in advection and vertical mixing modules, for instance), the result of the CPE-parallelized code could be deviated with the result produced by the MPE-only in the last 5 digits in double precision.

**Table 8** gives an example for which the deviated digits of the global mean SST is moved toward the left as the model integration forwards in the CPE-parallelized version of POP2 compared to its MPE-only version.

We also created a four-member ensemble (only four due to the constraint of computing resources)
of 1-year SW-CPE CESM-HR simulations (that differ by around-off level perturbation to the initial temperature) and computed the root mean square (RMS) of their differences on global SST fields.. We also compute the RMS of the differences between the SW-CPE and SW-MPE CESM-HR versions. These results are shown in **Fig. 10.** From **Fig. 10,** we found that the ensemble of the differences between the results of SW-CPE CESM-HR version and the SW-MPE CESM-HR
version was overall undistinguishable from the ensemble of the differences between either Intel(QNLM) and Intel(TAMU) machines or the SW-CPE perturbation runs, i.e., these simulations appear equivalent in terms of the SST.

We then applied the CESM-ECT test using the same procedure as for the SW-MPE-only version
described in section 3.2.2.  In particular, we used UF-CAM-ECT to compare three 9-timestep simulations with SW-CPE CESM-HR against the control ensemble from NCAR's Cheyenne machine. The overall ECT result for this comparison was a "fail", with nearly all of the 50 PCs tagged as failing.   This result, while concerning, requires further investigation for several reasons. First, while the CESM-ECT test has been very thoroughly tested at lower resolutions (e.g., 1
degree), this is its first application to a high-resolution run, and perhaps an adjustment to the test procedure could be required for finer grids.  Second, it is possible for CESM-ECT to legitimately fail at 9 timesteps due to a statistical difference at that timestep, while then issue a pass for the same simulation when evaluating the 1-year annual average, meaning that difference is not significant over the longer term.  (An example of such a scenario is changing the type of random
number generator as detailed in Milroy et al, 2018.)  Therefore, creating a large ensemble of 1-year high resolution simulations is an important next step.  Unfortunately, due to the computational cost required, this task remains future work.  The CESM-ECT is a sensitive test, and it will be challenging to determine the reason for failure even with the current tools available (Milroy et al. 2019).  However, in the meantime, the results of the multi-century control simulation (as discussed
in Section 4) continue to look quite promising with no noticeable issues.

### 3.7 The current CPE-parallelized version of CESM-HR

Following the refactoring and optimizing procedures for CAM5 and POP2, and other optimization strategies presented in **Fig. 9**, the optimization of CESM-HR on the Sunway machine resulted in a
simulation rate of 3.4 SYPD at the end of June, 2019, as shown in **Fig. 9**. There are 3 important stages that are worth mentioning in this optimization process: 1) at the end of 2018, based on MPE-





scaling and load balancing, using 48,000 CGs, after the CPE parallelization for CAM5 dynamic
core and physics modules, the model can have a 1.2 SYPD integration speed; 2) with the CPE-
parallelization for POP2 hot spot subroutines and further optimization on CAM5 dynamic core
communication and CAM5 load balancing, the model gains a 2.4 SYPD integration speed, at the
end of March, 2019; 3) after the optimization on malloc function and sea-ice initialization, as well
as further scaling and load-balancing with 65,000 CGs, the model reaches the current integration
speed of 3.4 SYPD (**Table 9**).

Currently, the CPE-parallelized CESM-HR has finished the PI_CTL simulation on the Sunway
machine over 400 years. While more diagnostics will be presented in the follow-up studies, the
next section will give some fundamental pictures of the Sunway machine integration results.

**4 The PI_CTL simulation of CESM-HR on the Sunway TaihuLight**

The first important piece of production simulations with the CESM-HR is to complete a 500-yr
pre-industrial control (PI_CTL) integration. We continue the PI_CTL simulation starting from the
initial conditions of January 1, 0021, which has been done on the Intel(TAMU) machine (for 20
years). Currently, another 400 years have been done and the model integration keeps continuing
with a stable rate.

**4.1 The feature of radiative balance at the top of atmosphere**

As discussed in section 3.2d, once the model includes CPE-parallelization, the CPE loop
computation carried out by c-language and its compiler could introduce a very small perturbation
into the model integration. Unlike a traditional model computed on a homogeneous multi-core Intel
machine that is a self-closed system in terms of machine precision (reproduceable 16 digits in
double precision, for instance), the Sunway CESM-HR computed in a hybrid mode in which the
MPE major task (in fortran-90) manages the CPE sub-tasks (in c-language) appears a frequently-
vibrating balance at the top of atmosphere (TOA) in such a frequently-perturbed system has a little
different behaviour from that of a self-closed system (see **Fig. 11**). In the self-closed model system
on the multi-core Intel machine (TAMU's Fronterra in this case), from an imbalanced default
initial state (1.2 Watts/m$^2$ of the difference between short-wave in and long-wave out) (green dot in
**Fig. 11**), the model quickly reaches the nearly-balanced states (.07 Watts/m$^2$ of in-out mean
difference) (red dot in **Fig. 11**) by roughly 10 years and then remain in oscillations with a small
amplitude of +-0.1 Watts/m$^2$. However, in the heterogeneous many-core Sunway system with a
frequently-perturbed hybrid mode, the time by which the model integration reaches a quasi-
equilibrium is much longer (about 60 years) and the oscillation amplitude is much bigger (+-0.6
Watts/m$^2$).

**4.2 Model bias**

The SST bias with the last 100-yr data as well as the bias of the global mean ocean temperature
and salinity against the climatological data are shown in **Figs. 12 and 13**. Compared to the
simulation results of coarse-resolution coupled model (Delworth et al., 2006; Collins et al., 2006),





the bias of ocean temperature and salinity simulated by CESM-HR is reduced by roughly 10%. As expected, the major model error regions by order are the north Atlantic, the north Pacific and the Southern Ocean Antarctic circumpolar current area.

**5 Summary and discussions**

Science advancement and societal needs require Earth system modelling with higher and higher
resolution which has a tremendous demanding of computing power. However, currently the semi-conductor technology gradually approaches its physical and heat limits, and recent supercomputers have adopted major architectural changes to continue increasing the performance through more power-efficient (faster but greener) heterogeneous many-core systems. The Sunway TaihuLight supercomputing platform is an outstanding example of heterogeneous many-core systems,
consisting of Management Processing Element (MPE) and Computing Processing Element (CPE) inside one processor. Adapting the legacy program and enabling the models onto such a new supercomputer is challenging but of great value for the global community. The Community Earth System Model with a high resolution of 10 km ocean and 25 km atmosphere (CESM-HR) has been successfully enabled and optimized on the Sunway TaihuLight, with millions of lines of legacy
codes being parallelized and redesigned on CPEs by a large group in iHESP (International Laboratory for High-Resolution Earth System Prediction) – a major international collaboration among QNLM (Qingdao Pilot National Laboratory for Marine Science and Technology), TAMU (Taxes A & M University) and NCAR. The current CPE-parallelized version has reached a simulation rate of 3.4 model years per wall clock day and has completed an unprecedented long
high-resolution simulation (400 years) of the pre-industrial climate. With further strategies on deeper refactoring and optimizing on computing hot spots, it is expected that the CPE-parallelized CESM-HR has an Intel-equivalent even Intel-beyond work efficiency.
The heterogeneous many-core systems have a new architecture and therefore require new programming strategies. Our current efforts focus on adapting the legacy codes being mostly
designed and developed for traditional homogeneous multi-core processors onto the new architecture machine without changing the significant algorithm design. Theoretically, the computational capacity of a Sunway core group (1 MPE+64 CPEs) with a dominant frequency of 1.45 GHz is 35.4 times of that of a traditional homogeneous core with a dominant frequency of 2.66 GHz. Due to the small CPE LDM (only 64KB in each CPE, plus needing to be managed by
the users) and the cost of frequent communication between MPE and CPEs, the current CPE optimization has constraints that relate to both the architecture and the existing software. In the follow-up studies, we may significantly redesign algorithms based on the new machine architecture fully considering the nature of CPE parallelization with a small user-controlled cache to minimize the communication cost. Moreover, since Earth system modelling with higher resolution is one of
major demands for supercomputing capacity and has its own scientific algorithm characteristics, in the new machine design, considering Earth system modelling-oriented architecture or partial functions may advance the development of both supercomputer and Earth system modelling. Although the focus of this paper is on the heterogeneous many-core Sunway TaihuLight supercomputing system, we believe many of the refactoring and optimizing processes detailed in
the paper can also be useful to the design of code porting and optimization strategies on other





heterogeneous many-core systems. We also acknowledge that the verification of correctness of recoding on a heterogeneous many-core supercomputing platform is very challenging, and we plan follow-up studies on using CESM-ECT for high-resolution simulations in general and its specific application to the current CPE optimization. Given the rapid development of heterogeneous many-
core supercomputing platforms, evaluating high resolution code refactoring efforts is quite important to the community.

**6 Code availability**

The Sunway version of the model is available at ZENODO via
https://doi.org/10.5281/zenodo.3637771.

**7 Author contributions**

LW, SZ, PC, and GD initiated this research, proposed most ideas, and co-led paper writing.
SZ, HF led the process of development & research.
YL, HW, YZ, XD, WW, LW, YZ, HM, KX, PX, LG, ZL, SW, YC, HY, and SS were responsible for code development, performance testing, and experimental evaluation.
GY, ZW, GP, YG participated in organization of working resources
HF, YL, WW, and LG were responsible for code release and technical support.
All authors contributed to the improvement of ideas, software testing, experimental evaluation, and paper writing/proofreading.

**8 Competing interests**

The authors declare that they have no conflict of interest.

**Acknowledgments**

The research is supported by the Key R & D program of Ministry of Science and Technology of China (2017YFC1404100, 2017YFC1404102), and Chinese NFS projects 41775100 and 41830964. Finally, this research is completed through the International Laboratory for High Resolution
Earth System Prediction (iHESP) – a collaboration among QNLM, TAMU and NCAR.

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





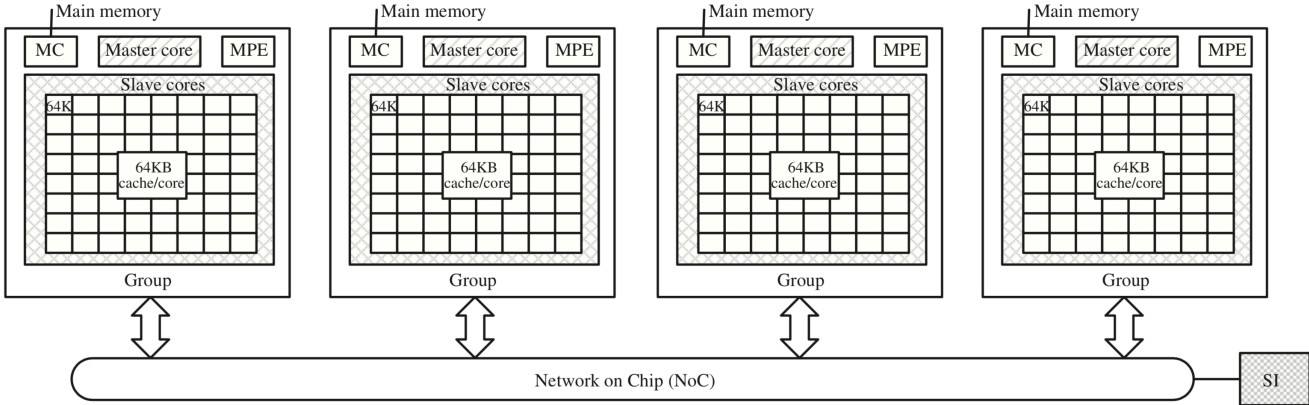

**Figure 1: A schematic illustration of the general architecture of the Sunway SW26010 CPU. Each CPU consists of 4 Core Groups, and each Core Group includes a Memory Controller, a Master Core (i.e. MPE - management processing element) and 64 Slave Cores (i.e. CPEs - computing processing elements), each of which has a 64-KB scratchpad fast memory, called LDM (local data memory). 4 Core Groups are linked together by the Network on Chip, and the whole CPU is linked with other CPUs by the System Interface (SI) network.**


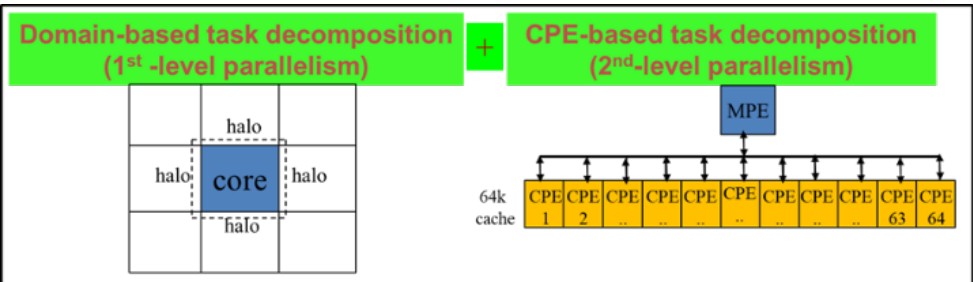

**Figure 2: Illustration of the 2nd level parallelism of CPEs-based task decomposition required on the Sunway heterogeneous many-core machine, additional to the 1st level parallelism of domain-based task decomposition among core groups, the MPI parallelism as in the homogeneous multi-core system.**






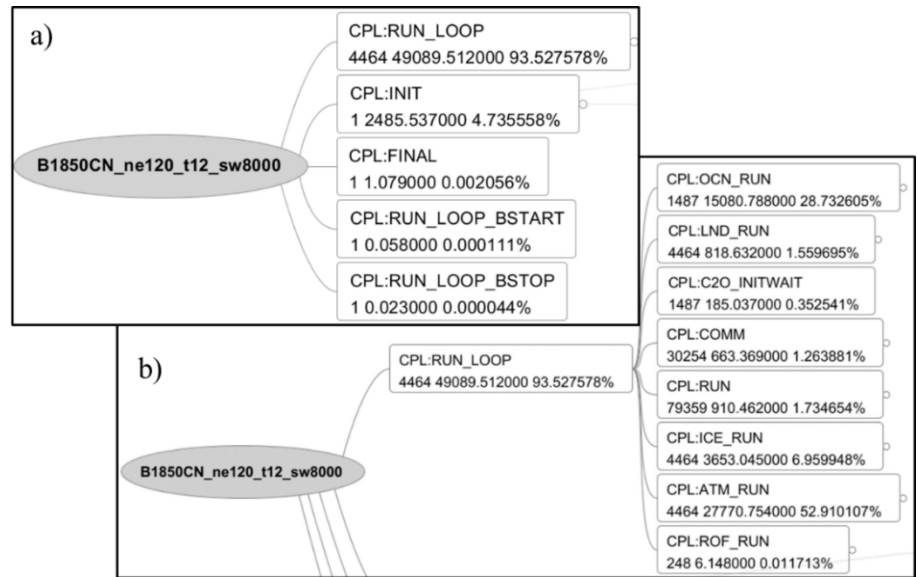

**Figure 3:** An example of tracking computing hot spots in major components of the 10kmOcn+25kmAtm high-resolution CESM for the pre-industrial control experiment with the 1850 fixed-year radiative forcing using 8000 core-groups on the Sunway machine (B1850CN_ne120_t12_sw8000) for one month integration: a) the total number of calls (in integer, 4464 calls of RUN_LOOP's in this experiment, for instance), the total seconds of computing time (in real, 49089.512 seconds for the 4464 calls of RUN_LOOP's, for instance) and projected percentage (93.528% for the 4464 calls of RUN_LOOP's, for instance) in 5 items of the main driver; b) further partitions of the 93.528% RUN_LOOP in 8 model components.

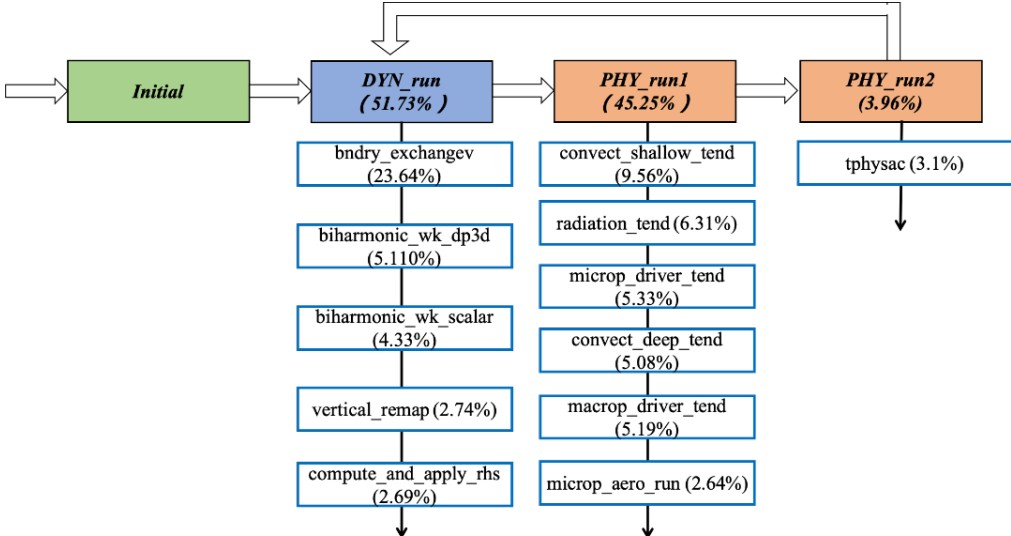

**Figure 4:** The major consumers of run time in the CAM5 model. The profiling is performed with a 25-km CAM5 run using 29,000 MPI processes.





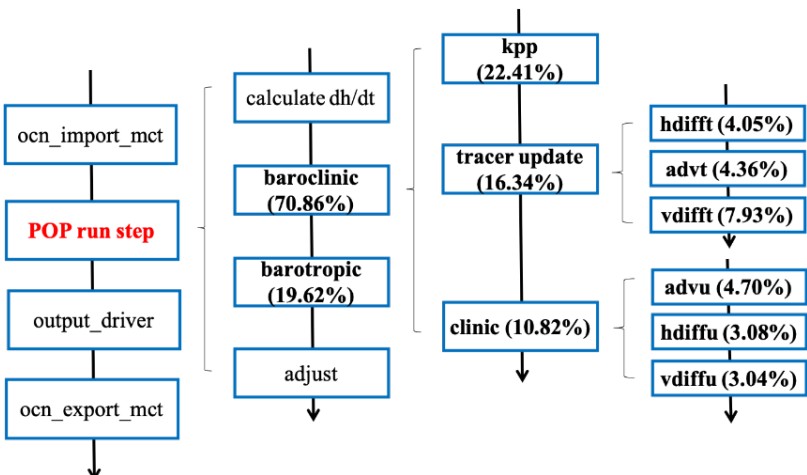

**Figure 5: The major consumers of run time in the POP2 model. The profiling is performed with a 10-km POP2 run using 18,300 MPI processes.**



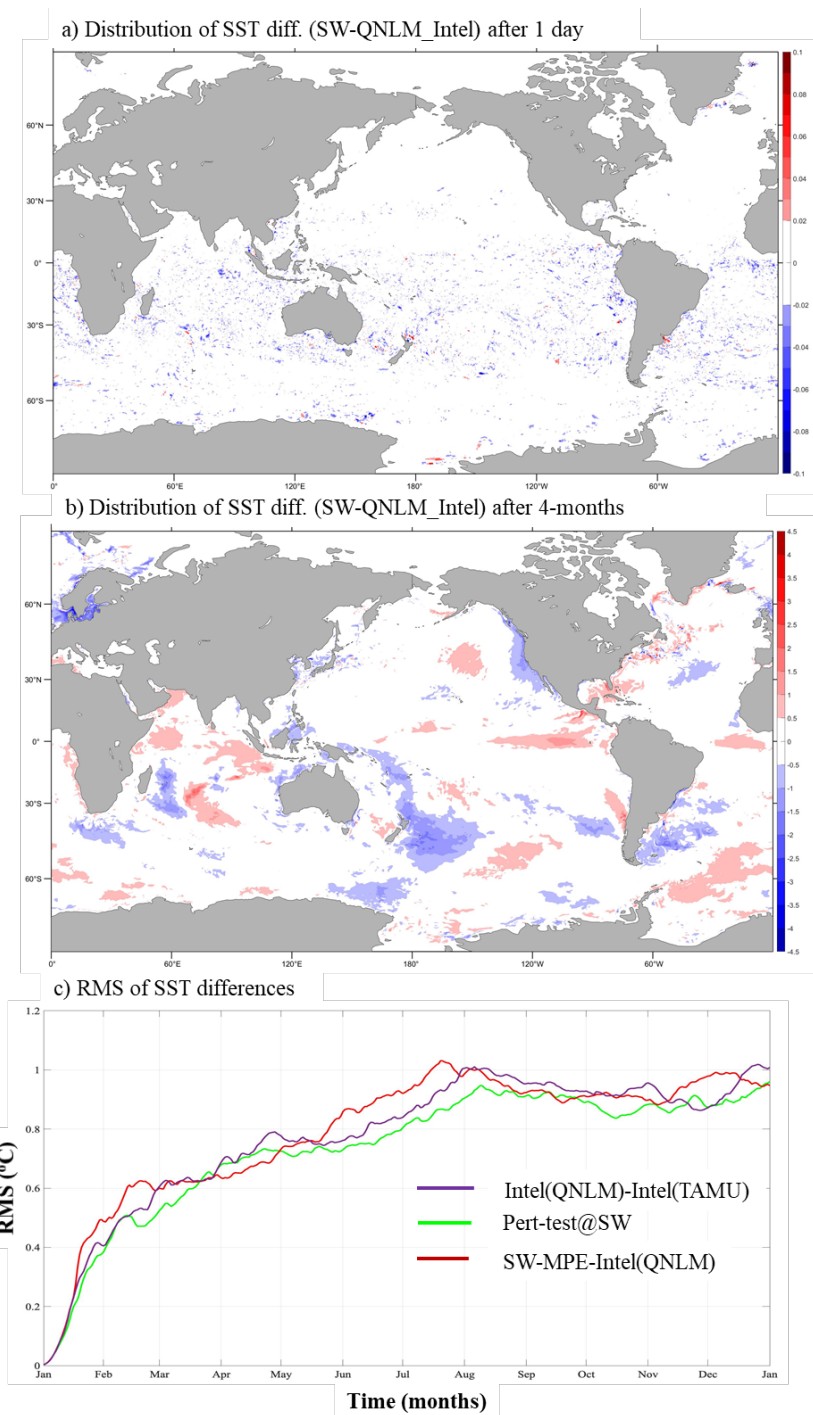

**Figure 6: The distribution of differences of sea surface temperatures produced by the Sunway machine and Intel machine at TAMU after a) 1-day, b) 1-yr integration, and c) the time series of root mean square of the differences between different machines and perturbation test.**





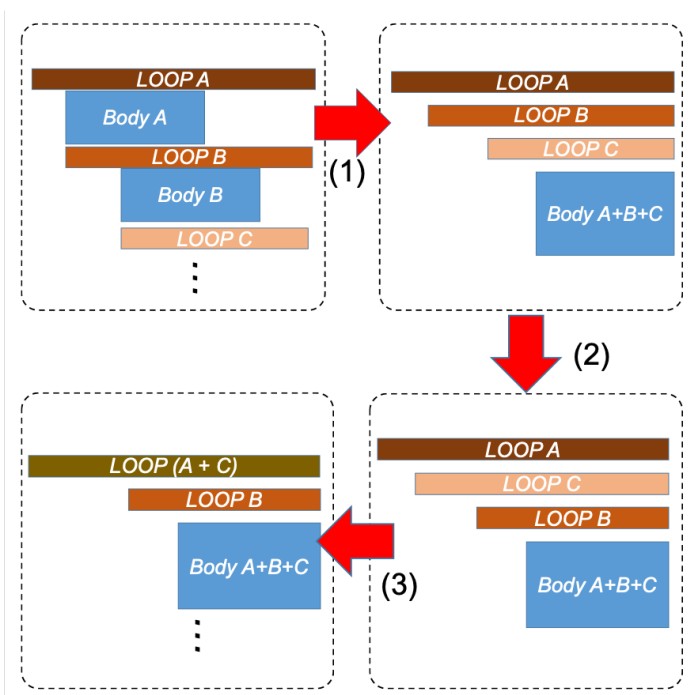

**Figure 7: Typical loop transformations performed to achieve the most suitable loop bodies that fit the number of parallel cores and the size of fast buffer in each core of the SW26010 many-core processor. (1) Aggregation of the loop body into the most inner loop. (2) Interchange of loop B and C. (3) Merge of loop A and C.**

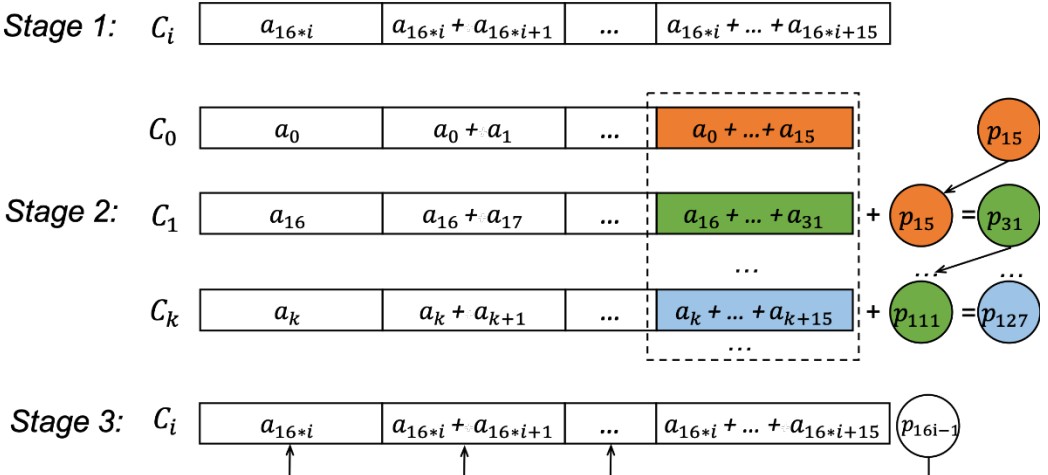

**Figure 8: Register communication-based parallelization of dependent loops: an example of computing the geopotential height at 128 different layers, parallelized as 8 CPE threads.**



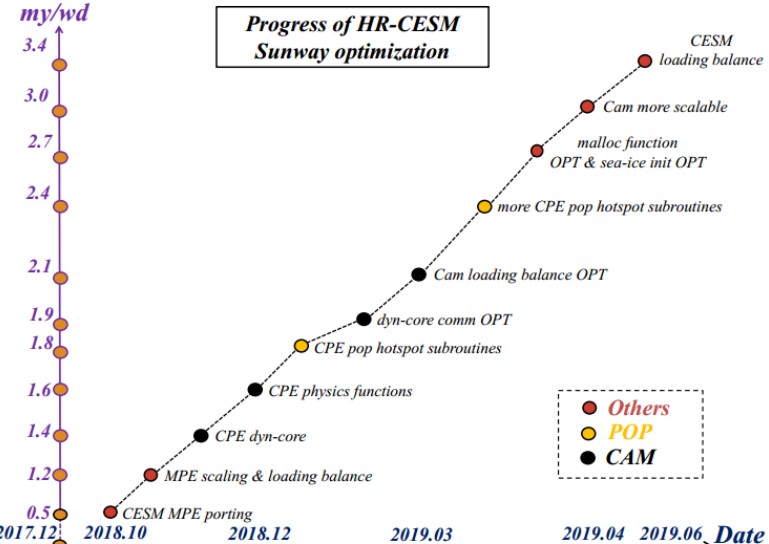

**Figure 9: The schedule of CESM-HR optimization on the Sunway TaihuLight.**


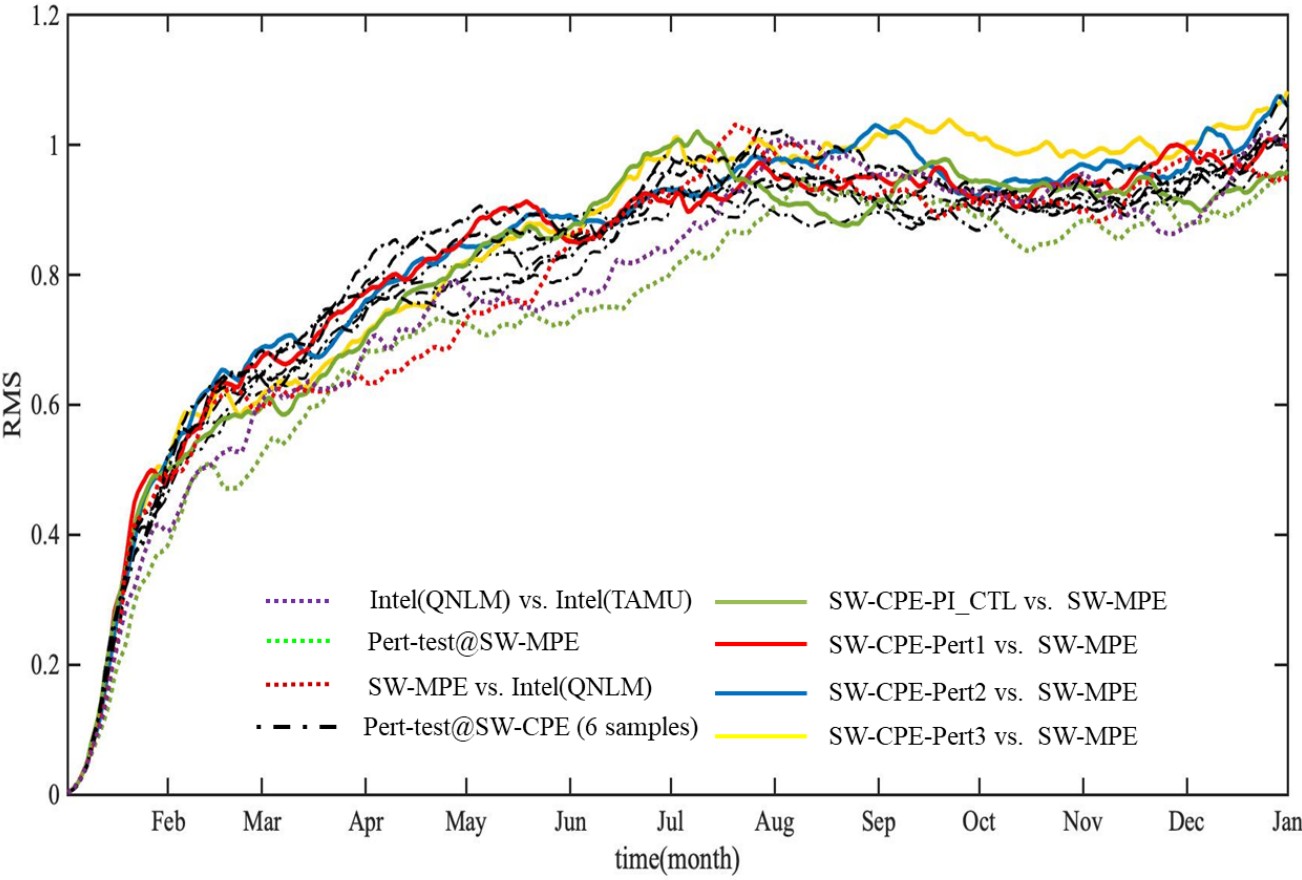

**Figure 10: Timeseries of the root mean squared (RMS) global SST differences between each of 3 perturbation and PI_CTL simulations of the CESM-HR SW-CPE version and the CESM-HR SW-MPE simulation (4 colour-solid lines), and any two of 4 CESM-HR SW-CPE simulations (6 black-dashed-dotted lines). 3 timeseries of RMS global SST differences shown in Fig. 6 between the Intel(QNLM) and Intel(TAMU) (pink-dotted), and two perturbed SW-MPE simulations (green-dotted) as well as the SW-MPE and Intel(QNLM) (red-dotted) are also plotted as the reference.**

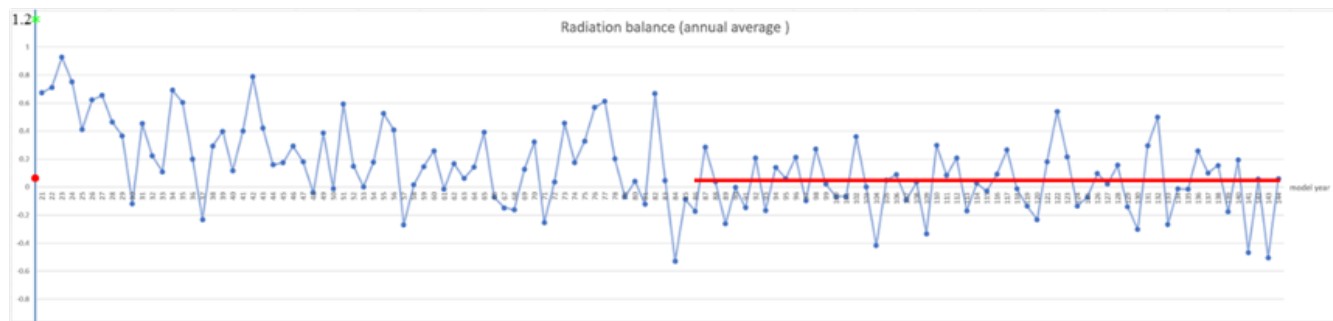





**Figure 11: Time series of annual mean global average radiation balance at the top of the atmosphere (difference of short-wave radiation in and long-wave radiation out). The red line marks the time average over yr-85 to yr 142 as a value of 0.05 Watts/m2. The green dot (1.2 Watts/m2) marks the initial value when the model starts from the default initial conditions while the red dot marks the last 5-yr mean value (.07 Watts/m2) after the model finishes 20-yr integration on the TAMU Intel machine.**


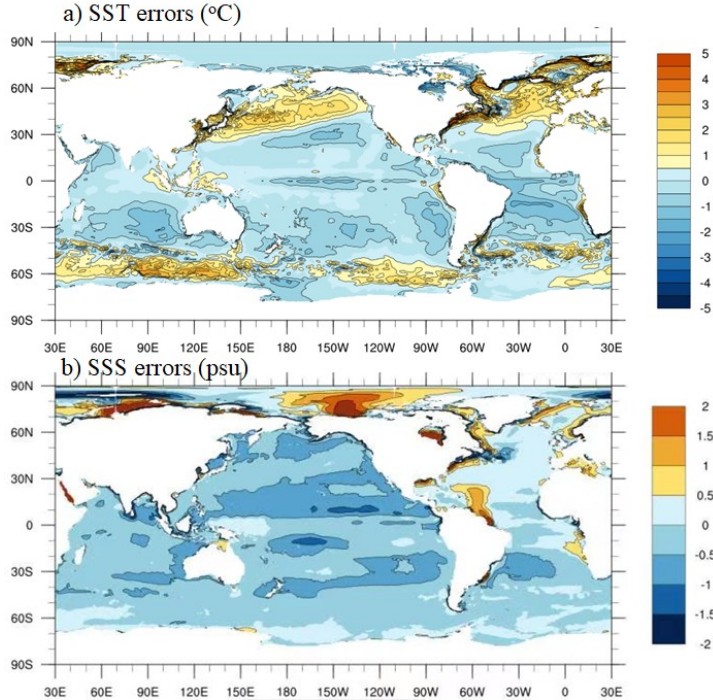


**Figure 12: The spatial distribution of time mean errors of a) SST and b) SSS simulated by the CPE-parallelized CESM-HR on the Sunway machine.**





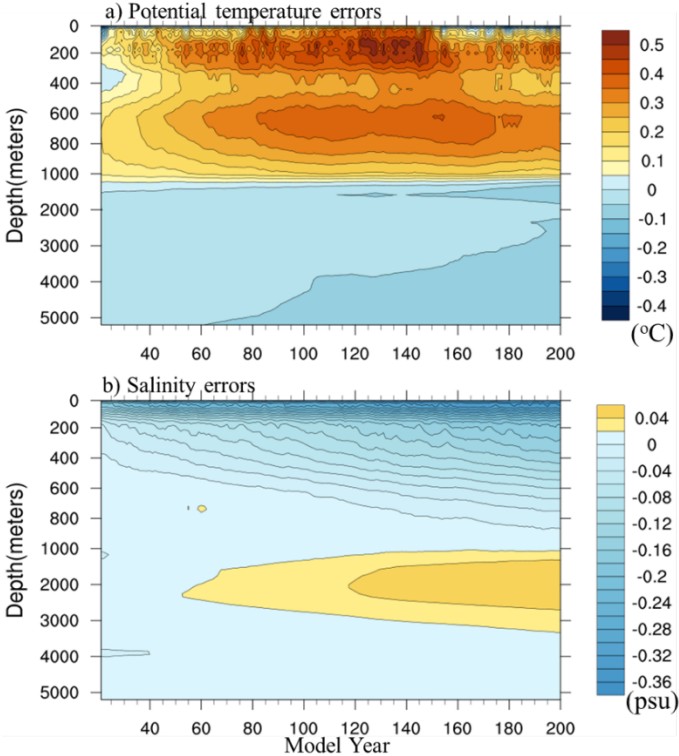

**Figure 13: Time series of global averaged errors of ocean a) temperature and b) salinity simulated by the CPE-parallelized CESM-HR on the Sunway machine against the climatological data.**

**Table 1: The list of diagnostic software tools in the Sunway system**



| Tool Name | Main Functions |
|---|---|
| SWLU | An MPE debug and profiling tool. The tool prints stack backtrace when getting a Linux signal, and uses sampling and backtrace to achieve a profiling result. |
| LWPF | A profiling tool for the CPE. The tool uses performance counters to measure the performance of CPEs. |
| SPC | A CPE debug tool. The tool prints error messages when crashed unexpectedly. |
| MALLOC_CHECKER | An memory leak detection tool. The tool wraps malloc function calls to check memory allocations and memory leaks. |
| SW5FCC | A compilation tool for reducing the OpenACC compilation time. The original swacc compiler takes a long time in JVM startup. The tool uses regular expression to exclude code files which are not dependent on swacc, and to process these files using mpicc directly. |
| SW5C | A compilation tool for long jumps in the CPEs. To get callee address, the original method needs to access the gp-based address which is not stored in LDM, and is slow to read. Considering that all function calls are static in CPEs, the tool uses constant splicing to calculate the address instead of reading the address stored in DDR memory. |
| GPTL | A tool for profiling the performance of parallel and serial applications on Sunway processors. |


Table 2:  Power Efficiency of Some Major Systems

| System | Rmax (PFlops) | Power Efficiency GFlops/Watts |
|---|---|---|
| Summit | 148.6 | 14 |
| TaihuLight | 93 | 6 |
| Tianhe-2A | 61 | 3 |
| Titan | 17.6 | 2.14 |
| Cheyenne Supercomputer | 1.3 | 0.86 |
| K Computer | 10 | 0.83 |





**Table 3: Timing of the CESM-HR on Sunway MPEs**

| Model Component | Timing (mons/wday) | Speeding Rate t(MPE-only)/t(MPE+CPE) |
|---|---|---|
| CAM | 18.5 | 1 (no CPE) |
| POP | 13.5 | 1 (no CPE) |
| CESM w/o IO | 13 | 1 (no CPE) |
| CESM with IO | 12 | 1 (no CPE) |


**Table 4: Results of the ECT tests for the Sunway MPE-only CESM-HR version with different compiling optimization options,mons/wday refers to Months/Wall O'clock.**

| Compiler optimization option | Number of PC scores failing at least two runs | CESM-ECT result |
|---|---|---|
| -O1 | 0 | PASS |
| -O2 | 3 | FAIL |
| -O3 | 1 | PASS |

**Table 5: List of refactoring and optimizing schemes for CAM5 physics modules with significant computing hot spots**





| Module Name | Functionality | CPE optimization schemes | Optimization rate of t(MPE)/t(CPE) with 18300 processes |
|---|---|---|---|
| rrtmg_sw_spcvmc.f90 | For shortwave radiation, containing spectral loop to compute the shortwave radiative fluxes, using the two-stream method of H. | 1) Transformation of independent loops;<br>2) Data reversing for vectorization;<br>3) Athread-based redesign of the code. | 8.9 |
| wetdep.F90 | wet deposition routines for both aerosols and gas phase constituents | 1) Optimizing of DMA;<br>2) Tiling of data;<br>3) Fast and LDM-saved math library. | 8.7 |
| convect_deep.F90 | CAM5 interface to several deep convection interfaces | 1) Optimizing of DMA;<br>2) Tiling of data;<br>3) Fast and LDM-saved math library;<br>4) Register communication for reduce operation. | 8.6 |
| cldwat2m_macro.F90 | CAM5 Interface for Cloud Macrophysics | 1) Optimizing of DMA;<br>2) Tiling of data;<br>3) Fast and LDM-saved math library. | 8.6 |
| modal_aero_wateruptake.F90 | computing aerosol wet radius | 1) Transformation of independent loops;<br>2) Taking control logics out of loops;<br>3) Optimizing of DMA.<br>4) Fast and LDM-saved math library. | 6.7 |
| micro_mg1_0.F90 | microphysics routine for each timestep | 1) Optimizing of DMA;<br>2) Tiling of data;<br>3) Vertical layer partition;<br>4) Optimizing of DMA;<br>5) Fast and LDM-saved math | 6.4 |





| | | library;<br>6) Athread-based redesign of the code. | |
|---|---|---|---|
| modal_aer_opt.F90 | parameterizing aerosol coefficients using Chebychev polynomial | 1) Optimizing of DMA;<br>2) Fast and LDM-saved math library | 6.4 |
| rrtmg_lw_rad.F90 | Computing long-wave radiation | 1) Transformation of independent loops;<br>2) Optimizing of DMA. | 3.1 |

**Table 6: List of refactoring and optimizing schemes for POP2 computing hotspot**

| Module Name | Functionality | CPE optimization schemes | Optimization rate t(MPE)/t(CPE) 18300 processes |
|---|---|---|---|
| vmix_kpp.F90 | Computing vertical mixing coefficients for the KPP parameterization | 1) Transformation of independent loops;<br>2) Taking logics out of loops;<br>3) Optimizing of DMA;<br>4) Tiling of data;<br>5) Athread-based redesign of the code. | 7.5 |
| hmix_del4.F90 | Computing biharmonic horizontal diffusion for momentum and tracers | 1) Transformation of independent loops;<br>2) Grouping of CPEs;<br>3) Taking logics out of loops;<br>4) Tiling of data. | 6.4 |
| advection.F90 | Performing advection of momentum and tracer quantities | 1) Transformation of independent loops;<br>2) Grouping of CPEs;<br>3) Taking logics out of loops;<br>4) Tiling of data. | 5.5 |
| vertical_mix.F90 | Computing vertical mixing tendencies, implicit vertical mixing and convection | 1) Transformation of independent loops;<br>2) Partition of big loop blocks;<br>3) Grouping of CPEs; | 4.7 |





| | | 4) Taking logics out of loops;<br>5) Double buffer;<br>6) Tiling of data; | |
| baroclinic.F90 | Computing baroclinic velocities and tracer fields. | 1) Transformation of independent loops;<br>2) Grouping of CPEs;<br>3) Function inline. | 3.4 |
| POP_SolversMod.F90 | elliptic solver for surface pressure in the barotropic mode | 1) Transformation of independent loops;<br>2) Taking logics out of loops. | 3.0 |
| step_mod.F90 | Forwarding the model one timestep | 1) Transformation of independent loops;<br>2) Optimizing of DMA. | 2.2 |


**Table 7: List of refactoring and optimizing schemes for POP2 computing hot spots**





| Subroutine Module name | Computation properties | Complex function calls | Difference between MPE-only and CPE-parallelization |
|---|---|---|---|
| subroutine btropOperator (in POP_SolversMod.F90) | 2-dimensional array multiplication and addition | No function calls involved and no complex computation | 0 (bitwise identical) |
| subroutine step (in step_mod.F90) | 2-dimensional array multiplication and addition | sqrt function called frequently | $10^{-16}$ (different last digit in double precision) |
| subroutine aduv (in advection.F90) | 2-dimensional array multiplication with many times | multiple-level array multiplications involved | $10^{-16}$ for most of points but maximum up to $10^{-13}$ (different last 4 digits in double precision) |
| subroutine ddmix (in vmix_kpp.F90) | 3- or higher-dimensional array multiplication and addition | exp function called frequently | $10^{-16}$ for most of points but maximum up to $10^{-12}$ (different last 5 digits in double precision) |

**Table 8: Movement of the difference of global mean SST in digits caused by CPE- parallelization by the forwarding of POP2 integration**


| Integration time | Global mean SST values | |
|---|---|---|
| 1 days | MPE-only | 18.148852140527389 |
| | CPE-parallelization | 18.148852141288383 |
| 2 days | MPE-only | 18.126134619263823 |
| | CPE-parallelization | 18.126134512879307 |
| 3 days | MPE-only | 18.115670343667720 |
| | CPE-parallelization | 18.115671056635556 |
| 4 days | MPE-only | 18.111063495112525 |
| | CPE-parallelization | 18.111063420599363 |
| 5 days | MPE-only | 18.112453155666980 |
| | CPE-parallelization | 18.112453439747853 |





**Table 9: The current CPE parallelism speeding rate of the HR CESM configured on 65000 CGs**

| Module | second | second/mday | myears/wday |
|---|---|---|---|
| TOT | 898.099 | 68.974 | 3.43 |
| LND | 73.963 | 5.680 | 41.67 |
| ROF | 0.717 | 0.055 | 4298.73 |
| ICE | 115.923 | 8.903 | 26.59 |
| ATM | 479.456 | 36.822 | 6.43 |
| OCN | 680.434 | 52.257 | 4.53 |
| CPL | 275.232 | 21.138 | 11.20 |
| CPL COMM | 420.185 | 32.270 | 6.34 |