# Peer review of "Optimizing High-Resolution Community Earth System Model on a Heterogeneous Many-Core Supercomputing Platform"

_Geoscientific Model Development, 2020_

## Referee Comment (RC1) · Mark Govett (Referee) · 25 Mar 2020

General Comments:

This was a very well organized and written paper. The paper described efforts to port a large legacy, climate code to the Sunway TaihuLight system. The unique architecture of the Sunway processor was described which helps the reader understand some of the changes that were needed for the climate application to run efficiently. The original code was designed and run on Intel-based processors. Performance and baseline

scientific results were made prior to porting the code. The work described detailed efforts to optimize the code, while maintaining sufficient accuracy in the solutions. As the authors admit, determining an acceptable level of scientific accuracy is an ongoing process determined by many factors. With a complex scientific application, thorough testing and evaluation using multiple criteria is needed to build confidence in the solution.

Specific Comments:

The authors describe extensive efforts to optimize the code, which included many common techniques. You spent a lot of time optimizing for the Sunway processor, but did not apply that level of effort to the original code. Some of the changes could have been applied to original code, making the comparison of performance more fairly represented.

The work appears specifically designed to target a single system with a unique processor. Was performance portability considered as a factor in this effort? Could the modified code run on an Intel-based system and how did the results compare to the original code. Given the fine-grain nature of the parallelization, would GPUs or high core count CPU processors be a target for this work? Addressing performance-portabilty would make the impact of this work much greater than the results you achieved targeting a single system.

The performance impact of the different types of code optimizations you made were not described. This would be a useful way to determine the tradeoff between portability and performance. For example, specific optimizations described in the Stage 1-3 optimizations were closely aligned to the Sunway processor. How much performance benefit were there for each of the stages and was it applied to a large portion of the code?

Regarding porting the code to the TaihuLight system, your approach seemed to be first port the code to two Intel based systems (TAMU and QNLM). It was unclear why

you felt the need to port to both systems. Further, your comparisons were made after only 9 timesteps seemed arbitrary and perhaps no sufficient. Please include some justification. It was also unclear what fields were compared in the UF-CAM-ECT test. A summary of the relevant details from the paper would be useful here.

Are there references for the tools given in Table 1? Most of the tools listed were not referenced in the manuscript. They should be either introduced to the reader in the paper if they add value to the manuscript. For example, you could state how you used them and how it helped identify

Technical Corrections:

Line 103-105: Awkward sentence. Perhaps break into two sentences?

Line 140: Old reference (Govett, 2010) should be replaced with a more comprehensive paper (Govett, et al. 2015) in the Bulletin of the AMS: https://doi.org/10.1175/BAMS-D-15-00278.1

Lines 212-218: The authors don't describe what the speed and bandwidth of connections between super-nodes, within a cabinet, and between cabinets. This is essential in understanding the limitations of the Sunway TaihuLight system at scale.

Line 243: change "details Section 3.3" to: "detail in Section 3.3"

Line 245-250: unclear if CPE based parallelism is with MPI or something else??? Line 251: change "details" to "detail"

Lines 260-266: It appears that an MPI-based intelligent programming model would work here. The intelligence would be knowing when comms within a CPU task group or to an MPE are needed.

Lines 268-272: Regarding the power efficiency comparison in Table 2, is this a fair comparison? It seems like there would be size benefits favoring the larger systems especially in terms of infrastructure required no matter the size of the system.

[Figure]

Line 307: swlu was italicized but not introduced in section 2.1.2. It appears in capitals as SWLU in Table 1. .. Line 332: TAMU and QNLM are undefined (QNLM is defined on line 602, TASMU on line 603)

Line 341: It seems that such short runs are not sufficient. Did you make a similar test with more than nine time steps. There remains a high potential for variations to show up later n the simulation experiments

Optimizations were made to achieve 1 SYPD on the Sunway system. Did you attempt to incorporate these changes and optimization techniques back into the original model?

Line 372: Can you provide more details or analysis regarding why -O2 fails but -O3 passes How long were the runs made before comparisons were done?

---

## Referee Comment (RC2) · Carlos Osuna (Referee) · 11 May 2020

This paper provides a comprehensive description of the efforts to port large legacy code of the CESM model to the Sunway TaihuLight processors. The text gives a very detailed description of the improvements made to the model, the parallelization and optimization techniques employed as well as the programming models used (OpenACC and Athread). A pre-industrial simulation over 400 years is being performed and the main result is an optimization from 1 SYPD to 3.4 SYPD. The paper is well structured and provides comprehensive information about the model, and the experiments for

reproducibility. The software is open source and referenced within the paper.

General comments: the text is sometimes too dense and hard to follow. I would recommend to interleave the relevant tables and figures within the text, next to the references and the discussion. Otherwise it is hard to follow references to tables and figures that are placed at the end.

For a paper running on such a large system (65000CGs) , a scalability plot is missing.

The paper emphasizes in several places the energy consumption point of view and the advantage of hybrid architectures like TaihuLight. However there is no real data for this experiment presented, therefore there is no data that support some strong statements presented in the text.

Line 105: the argument is valid only for applications that maximize the FLOPs provided by a computer, which does not hold for weather and climate applications. Same happens with Table 2 which presents general specs measured for different machines. If 3.4 SYPD are achieved with 65000, while the benchmarks with ∼11000 Intel processors runs at ∼1SYPD I conclude that for the same SYPD, the TaihuLight requires 2 times more processors than the Intel system. How does the energy efficiency of the TaihuLight chip compares to the Intel? I encourage to backup the energy arguments with data from the experiments or make a more clear link.

The authors (very rightly) emphasize the large efforts required to port this large model to a new architecture. Here the reviewer is missing a more general discussion about the cost of these effort and performance portability. Was performance portability an important metric of this work? Considering that the TaihuLight is a unique system that didnt go into the market of supercomputing, what is the cost of such refactoring ? It would be interesting for the paper to provide a number for financial costs of the porting effort.

More specific comments:

The introduction is too long. It can be simplified and emphasize the contributions of this work. The related work part of the introduction is well written.

line 72: I dont think there is any major system in the list of supercomputers with FPGAs.

line: 158-159 This is not very correct. The authors already mentioned in the literature NIM that was ported to GPU and XeonPhi. Other models like ICON are GPU ready .

line 265: This is a very strong statement. It is true that the systems are very different. But from these differences is not obvious that a GPU system with multiple GPUs connected with a high-throughput low latency NVLink is not more suitable for scientific computations.

Section 2.2.2: as already mentioned, this refer to general specifications of the system which are for sure different than the FLOP/energy consumption of weather applications. I would consider providing experiment data or removing this section.

Section 3.1: It is hard to follow the text and what are the major improvement. Considering supporting with data, figures and simplifying the list.

---

## Author Comment (AC1) · 7 Jun 2020

Mark Govett (Referee) mark.w.govett@noaa.gov

General Comments: This was a very well organized and written paper. The paper described efforts to port a large legacy, climate code to the Sunway TaihuLight sys-

tem. The unique architecture of the Sunway processor was described which helps the reader understand some of the changes that were needed for the climate application to run efficiently. The original code was designed and run on Intel-based processors. Performance and baseline scientific results were made prior to porting the code. The work described detailed efforts to optimize the code, while maintaining sufficient accuracy in the solutions. As the authors admit, determining an acceptable level of scientific accuracy is an ongoing process determined by many factors. With a complex scientific application, thorough testing and evaluation using multiple criteria is needed to build confidence in the solution.

RE: Thank you for the thorough examination on our manuscript (MS) and constructive comments. We agree that all of the comments are very useful for us to improve the presentation of the MS, and we have fully addressed them in the revision.

What follows is a point-by-point reply to each comment.

Specific Comments:

(1) The authors describe extensive efforts to optimize the code, which included many common techniques. You spent a lot of time optimizing for the Sunway processor, but did not apply that level of effort to the original code. Some of the changes could have been applied to original code, making the comparison of performance more fairly represented.

RE: This is a good point that needs to be addressed in the MS. The work presented in this paper represents the first step of running CESM on the new architecture machine. To minimize the coding uncertainties, we keep the original CPU code unchanged for both accuracy verification and performance evaluation. It is true that many of the applied techniques can be generalized to CPU or even GPU architectures. That would be the focus of a later-stage study, which is to extract both general practices and tools that would help the transition from current multi-core CPUs to many-core accelerators. Redesigning some of the original algorithms is expected to further improve the efficiency.

Some clarifications and discussions are added in the revision. Please see L43-44; L125-126; L432-433; L532-533. Thanks a lot!

(2) The work appears specifically designed to target a single system with a unique processor. Was performance portability considered as a factor in this effort? Could the modified code run on an Intel-based system and how did the results compare to the original code. Given the fine-grained nature of the parallelization, would GPUs or high core count CPU processors be a target for this work? Addressing performance-portability would make the impact of this work much greater than the results you achieved targeting a single system.

RE: We agree that addressing the issue of general performance-portability is an important and interesting topic in using heterogeneous many-core HPC systems and should be further clarified and discussed more in the MS. In the revision we expanded the discussion of this topic. Please see L128-131; L370-373; L375; L387-388; L532-533; L540-543. Thanks a lot!

(3) The performance impact of the different types of code optimizations you made were not described. This would be a useful way to determine the tradeoff between portability and performance. For example, specific optimizations described in the Stage 1-3 optimizations were closely aligned to the Sunway processor. How much performance benefit were there for each of the stages and was it applied to a large portion of the code?

RE: Combined with comment (2), this is a very good comment addressing the general performance-portability issue. The description of performance impact of the different types of code optimizations is now added. Please see L370-373; L375; L387-388. The three-stage optimizations described in section are proposed to solve a specific prefix summation calculation using register communication. Our method is 27 times faster than the original code that has only a single loop. More discussions about its performance portability are added in the revision. Please see L370-373.

(4) Regarding porting the code to the TaihuLight system, your approach seemed to be first port the code to two Intel based systems (TAMU and QNLM). It was unclear why you felt the need to port to both systems. Further, your comparisons were made after only 9 timesteps seemed arbitrary and perhaps no sufficient. Please include some justification. It was also unclear what field were compared in the UF-CAM-ECT test. A summary of the relevant details from the paper would be useful here.

RE: Given the totally new architecture of the Sunway machine, to minimize the uncertainties of code porting, before we port the CESM-HR to the Sunway TaihuLight, we first ran and tested its correctness on the Intel multi-core supercomputing platforms available to iHESP, serving as the first benchmark. The justification and adjusted statement are made in the revision. Please see L284-287.

More descriptions and discussions about UF-CAM-ECT tools are added in the revision. Please see L293-295; L461-464. Thanks.

(5) Are there references for the tools given in Table 1? Most of the tools listed were not referenced in the manuscript. They should be either introduced to the reader in the paper if they add value to the manuscript. For example, you could state how you used them and how it helped identify

RE: The point is well taken! One more column describing the role of each tool in this project is now added in the Table 1. More reference information about Table 1 is added in the revision. Please see the new Table 1 and L193-194. Thanks.

Technical Corrections: Line 103-105: Awkward sentence. Perhaps break into two sentences?

RE: The entire sentence is removed in the revision. Thanks.

Line140 (104 in revised version): Old reference (Govett, 2010) should be replaced with a more comprehensive paper (Govett, et al. 2015) in the Bulletin of the AMS: https://doi.org/10.1175/BAMSD-15-00278.1

RE: Done. Please see L104; L615-617. Thanks.

Lines 212-218 (170-175): The authors don't describe what the speed and bandwidth of connections between super-nodes, within a cabinet, and between cabinets. This is essential in understanding the limitations of the Sunway TaihuLight system at scale.

RE: Good comment! The TaihuLight compute nodes are connected via a 2-level InfiniBand network. A single-switch with full bisection bandwidth connects all 256 nodes within a super-node, while a fat-tree with 1/4 of the full bisection bandwidth connects all super-nodes (as shown the attached Figure 1). Table II (as shown the attached Figure 2) shows measurements of bisection communication bandwidth at different levels of the system.

Such essential information is added in the revision. Please see L175-178. Thanks.

Line 243 (198): change "details Section 3.3" to: "detail in Section 3.3" Line 245-250 (200-205): unclear if CPE based parallelism is with MPI or something else??? Line 251 (207): change "details" to "detail"

RE: Line 198: Done; Line 207: Done. Line 245-250: The statements are revised for clarification. Please see L204-207.

Lines 260-266 (215-221): It appears that an MPI-based intelligent programming model would work here. The intelligence would be knowing when comes within a CPU task group or to an MPE are needed.

RE: Thanks for reminding the clarification. The statement is further clarified based on the comment. Please see L218-221.

Lines 268-272 (223-230): Regarding the power efficiency comparison in Table 2, is this a fair comparison? It seems like there would be size benefits favoring the larger systems especially in terms of infrastructure required no matter the size of the system.

RE: This is a good point! We all agree that the power efficiency is currently an issue

with rich uncertainties. More discussions on the uncertainties and future work direction are added in the revision. Please see L225-230. Thanks

Line 307 (257): swlu was italicized but not introduced in section 2.1.2. It appears in capitals as SWLU in Table 1.

RE: The tool name in Table 1 is modified to italicized, consistent with the context now. Thanks.

Line 332 (279): TAMU and QNLM are undefined (QNLM is defined on line 602, TASMU on line 603)

RE: In the revision, TAMU and QNLM are first defined at L116-117. Thanks.

Line 341 (288): It seems that such short runs are not sufficient. Did you make a similar test with more than nine time steps. There remains a high potential for variations to show up later in the simulation experiments Optimizations were made to achieve 1 SYPD on the Sunway system. Did you attempt to incorporate these changes and optimization techniques back into the original model?

RE: We also compare the results of 1-yr long runs in different perturbation scenarios (as shown in Fig. 10) to comprehend the integrations on Sunway machine. More discussions on the 9 timesteps are added (please see L460-463). To minimize the coding-caused uncertainties in the porting and optimizing process, at this stage, the current work doesn't apply any change to the original code. More statement and discussions are added in the revision. Please see L125-126. Thanks.

Line 372 (316): Can you provide more details or analysis regarding why -O2 fails but -O3 passes? How long were the runs made before comparisons were done?

RE: While more discussions on the metrics of the ECT tool are added in L460-463, the possible reason for which -O2 fails but -O3 passes is given in the revision. Please see L318-319; L460-463.

[Figure]

**Fig. 1.** TaihuLight Network System

**TABLE II**
**TaihuLight Specifications and Bandwidth Measured**

| Component | Configurations | BW measured (% peak) |
|---|---|---|
| MPE | 1.45 GHz, 32/256KB L1/L2 | DRAM 8.0 GB/s (80%) |
| CPE | 1.45 GHz, 64KB SPM | Reg. 630 GB/s (85%) |
| CG | 1 MPE + 64 CPEs | DRAM 28.98 GB/s (85%) |
| Node | 1 CPU (4 CGs), 4×8GB RAM | Net: 6.04 GB/s (89%) |
| Super Node | 256 nodes, FDR 56 Gbps IB | Bisection 1.5 TB/s (72%) |
| TaihuLight | 160 supernodes | Bisection 49.2 TB/s (68.6%) |
| Agg. memory | 1.3 PB | 4.6 PB/s (85%) |
| Storage | 5.2 PB (Online2) | 70 GB/s (54.35%) |

**Fig. 2.** TaihuLight Specifications and Bandwidth Measured

---

## Author Comment (AC2) · 7 Jun 2020

Carlos Osuna (Referee) carlos.osuna@meteoswiss.ch

This paper provides a comprehensive description of the efforts to port large legacy code of the CESM model to the Sunway TaihuLight processors. The text gives a very

detailed description of the improvements made to the model, the parallelization and optimization techniques employed as well as the programming models used (OpenACC and Athread). A pre-industrial simulation over 400 years is being performed and the main result is an optimization from 1 SYPD to 3.4 SYPD. The paper is well structured and provides comprehensive information about the model, and the experiments for reproducibility. The software is open source and referenced within the paper.

General comments: the text is sometimes too dense and hard to follow. (1) I would recommend to interleave the relevant tables and figures within the text, next to the references and the discussion. Otherwise it is hard to follow references to tables and figures that are placed at the end. (2) For a paper running on such a large system (65000CGs), a scalability plot is missing. (3) The paper emphasizes in several places the energy consumption point of view and the advantage of hybrid architectures like TaihuLight. However, there is no real data for this experiment presented, therefore there is no data that support some strong statements presented in the text.

RE: Thanks for the reviewer's thorough examination of our manuscript (MS) and positive comments. We all agree that your comments are very constructive for us to improve presentation of the MS, and all your major comments and other points have been fully addressed in the revision. Specifically, in the revision, (1) the Tables and Figs. are inserted into the relevant places; (2) a scalability plot is added; (3) the energy consumption statement is appropriately re-written and more discussions about the uncertainties of the current work on power efficiency are added.

The point-by-point replies are followed.

Line105 (89 in revised version): the argument is valid only for applications that maximize the FLOPs provided by a computer, which does not hold for weather and climate applications. Same happens with Table 2 which presents general specs measured for different machines. If 3.4 SYPD are achieved with 65000, while the benchmarks with 11000 Intel processors runs at 1 SYPD I conclude that for the same SYPD, the Taihu-

Light requires 2 times more processors than the Intel system. How does the energy efficiency of the TaihuLight chip compares to the Intel? I encourage to backup the energy arguments with data from the experiments or make a clearer link. The authors (very rightly) emphasize the large efforts required to port this large model to a new architecture. Here the reviewer is missing a more general discussion about the cost of these effort and performance portability. Was performance portability an important metric of this work? Considering that the TaihuLight is a unique system that did not go into the market of supercomputing, what is the cost of such refactoring? It would be interesting for the paper to provide a number for financial costs of the porting effort.

RE: We all agree that the power efficiency is currently an issue with rich uncertainties. The statement of power efficiency is modified, and more discussions on the uncertainties and future work direction are added in the revision. Please see L225-230. The discussions on the performance-portability issue are added in the revision. Please see L128-131; L370-374; L375; L387-388; L532-533; L540-543. Thanks a lot!

More specific comments:

The introduction is too long. It can be simplified and emphasize the contributions of this work. The related work part of the introduction is well written.

RE: The introduction is condensed, focusing on relevant project now. Please see the new introduction. Thanks.

line72: I don't think there is any major system in the list of supercomputers with FPGAs.

RE: The related statement is removed. Thanks.

line: 158-159 This is not very correct. The authors already mentioned in the literature NIM that was ported to GPU and XeonPhi. Other models like ICON are GPU ready. RE: The sentence has been removed in the revision. Thanks.

line 265 (218): This is a very strong statement. It is true that the systems are very different. But from these differences is not obvious that a GPU system with multiple

GPUs connected with a high-throughput low latency NVLink is not more suitable for scientific computations.

RE: The statement has been re-written in a more appropriate way. Please see L218-221.

Section 2.2.2: as already mentioned, this refer to general specifications of the system which are for sure different than the FLOP/energy consumption of weather applications. I would consider providing experiment data or removing this section.

RE: Agree that energy consumption is a complex issue and at the current stage, the Sunway machine has not shown any advantage on power efficiency. We may pursue really-greener utilization in the future when more experiences and optimization skills can make the computation with much higher efficiency. Please see L226-230.

Section 3.1: It is hard to follow the text and what are the major improvement. Considering supporting with data, figures and simplifying the list.

RE: The major improvement of CESM1.3-beta17_sehires38 has been reorganized. Please see the new Section 3.1. Thanks.

---

## Author Comment (AC3) · 12 Jun 2020

The Tables and Figs. are inserted into the relevant places in the marked-up version only, not the updated MS, as it is required to put them at the end.

---

## Editor Decision (ED1)

Mark Govett (Referee) mark.w.govett@noaa.gov

General Comments: This was a very well organized and written paper. The paper described efforts to port a large legacy, climate code to the Sunway TaihuLight system. The unique architecture of the Sunway processor was described which helps the reader understand some of the changes that were needed for the climate application to run efficiently. The original code was designed and run on Intel-based processors. Performance and baseline scientific results were made prior to porting the code. The work described detailed efforts to optimize the code, while maintaining sufficient accuracy in the solutions. As the authors admit, determining an acceptable level of scientific accuracy is an ongoing process determined by many factors. With a complex scientific application, thorough testing and evaluation using multiple criteria is needed to build confidence in the solution.

RE: Thank you for the thorough examination on our manuscript (MS) and constructive comments. We agree that all of the comments are very useful for us to improve the presentation of the MS, and we have fully addressed them in the revision.

What follows is a point-by-point reply to each comment.

Specific Comments:
(1) The authors describe extensive efforts to optimize the code, which included many common techniques. You spent a lot of time optimizing for the Sunway processor, but did not apply that level of effort to the original code. Some of the changes could have been applied to original code, making the comparison of performance more fairly represented.

RE: This is a good point that needs to be addressed in the MS. The work presented in this paper represents the first step of running CESM on the new architecture machine. To minimize the coding uncertainties, we keep the original CPU code unchanged for both accuracy verification and performance evaluation. It is true that many of the applied techniques can be generalized to CPU or even GPU architectures. That would be the focus of a later-stage study, which is to extract both general practices and tools that would help the transition from current multi-core CPUs to many-core accelerators. Redesigning some of the original algorithms is expected to further improve the efficiency. Some clarifications and discussions are added in the revision. Please see L43-44; L125-126; L432-433; L532-

533. Thanks a lot!

(2) The work appears specifically designed to target a single system with a unique processor. Was performance portability considered as a factor in this effort? Could the modified code run on an Intel-based system and how did the results compare to the original code. Given the fine-grain nature of the parallelization, would GPUs or high core count CPU processors be a target for this work? Addressing performance-portability would make the impact of this work much greater than the results you achieved targeting a single system.

RE: We agree that addressing the issue of general performance-portability is an important and interesting topic in using heterogeneous many-core HPC systems and should be further clarified and discussed more in the MS. In the revision we expanded the discussion of this topic. Please see L128-131; L370-373; L375; L387-388; L532-533; L540-543. Thanks a lot!

(3) The performance impact of the different types of code optimizations you made were not described. This would be a useful way to determine the tradeoff between portability and performance. For example, specific optimizations described in the Stage 1-3 optimizations were closely aligned to the Sunway processor. How much performance benefit were there for each of the stages and was it applied to a large portion of the code?

RE: Combined with comment (2), this is a very good comment addressing the general performance-portability issue. The description of performance impact of the different types of code optimizations is now added. Please see L370-373; L375; L387-388. The three-stage optimizations described in section are proposed to solve a specific prefix summation calculation using register communication. Our method is 27 times faster than the original code that has only a single loop. More discussions about its performance portability are added in the revision. Please see L370-373.

(4) Regarding porting the code to the TaihuLight system, your approach seemed to be first port the code to two Intel based systems (TAMU and QNLM). It was unclear why you felt the need to port to both systems. Further, your comparisons were made after only 9 timesteps seemed arbitrary and perhaps no sufficient. Please include some justification. It was also unclear what fields were compared in the UF-CAM-ECT test. A summary of the relevant details from the paper would be useful here.

RE: Given the totally new architecture of the Sunway machine, to minimize the uncertainties of code porting, before we port the CESM-HR to the Sunway TaihuLight, we first ran and tested its correctness on the Intel multi-core supercomputing platforms available to iHESP, serving as the first benchmark. The justification and adjusted statement are made in the revision. Please see L284-287.

More descriptions and discussions about UF-CAM-ECT tools are added in the revision. Please see L293-295; L461-464. Thanks.

(5) Are there references for the tools given in Table 1? Most of the tools listed were not referenced in the manuscript. They should be either introduced to the reader in the paper if they add value to the manuscript. For example, you could state how you used them and how it helped identify

RE: The point is well taken! One more column describing the role of each tool in this project is now added in the Table 1. More reference information about Table 1 is added in the revision. Please see the new Table 1 and L193-194. Thanks.

SV3: OK for Table 1 but this makes me realize that your section 6 on Data availability and section 7 and 8 have disappeared. Your Data availability section has to include the CESM code AND all tools.

Technical Corrections:

Line 103-105: Awkward sentence. Perhaps break into two sentences?

RE: The entire sentence is removed in the revision. Thanks.

OK

Line140 (104 in revised MS): Old reference (Govett, 2010) should be replaced with a more comprehensive paper (Govett, et al. 2015) in the Bulletin of the AMS: https://doi.org/10.1175/BAMSD-15-00278.1

RE: Done. Please see L104; L615-617. Thanks.

OK but lines 610-613, not 615-617

Lines 212-218 (170-175): The authors don't describe what the speed and bandwidth of connections between super-nodes, within a cabinet, and between cabinets. This is essential in understanding the limitations of the Sunway TaihuLight system at scale.

Text

RE: Good comment! The TaihuLight compute nodes are connected via a 2-level InfiniBand network. A single-switch with full bisection bandwidth connects all 256 nodes within a super-node, while a fat-tree with 1/4 of the full bisection bandwidth connects all super-nodes (as shown the following Figure). Table II shows measurements of bisection communication bandwidth at different levels of the system.

[Figure]

**TABLE II**
TAIHULIGHT SPECIFICATIONS AND BANDWIDTH MEASURED

| Component | Configurations | BW measured (% peak) |
|---|---|---|
| MPE | 1.45 GHz, 32/256KB L1/L2 | DRAM 8.0 GB/s (80%) |
| CPE | 1.45 GHz, 64KB SPM | Reg. 630 GB/s (85%) |
| CG | 1 MPE + 64 CPEs | DRAM 28.98 GB/s (85%) |
| Node | 1 CPU (4 CGs), 4×8GB RAM | Net. 6.04 GB/s (89%) |
| Super Node | 256 nodes, FDR 56 Gbps IB | Bisection 1.5 TB/s (72%) |
| TaihuLight | 160 supernodes | Bisection 49.2 TB/s (68.6%) |
| Agg. memory | 1.3 PB | 4.6 PB/s (85%) |
| Storage | 5.2 PB (Online2) | 70 GB/s (54.35%) |

Such essential information is added in the revision. Please see L175-178. Thanks.

OK but lines 173-175, not 175-178.

Line 243 (198): change "details Section 3.3" to: "detail in Section 3.3" Line 245-250 (200-205): unclear if CPE based parallelism is with MPI or something else??? Line 251 (207): change "details" to "detail"

RE: Line 198: Done; Line 207: Done.
Line 245-250: The statements are revised for clarification. Please see L204-207.

SV4: OK for "detail in Section 3.3" and "detail" but I don't see any revision at lines 204-207 regarding the CPE based parallelism.

Lines 260-266 (215-221): It appears that an MPI-based intelligent programming model would work here. The intelligence would be knowing when comms within a CPU task group or to an MPE are needed.

RE: Thanks for reminding the clarification. The statement is further clarified based on the comment. Please see L218-221. OK but the modification is at lines 216-218

SV5: 2.2.1: Add an "s" to "seem" in "… GFlops/Watts, seem to be greener …"

Lines 268-272 (223-230): Regarding the power efficiency comparison in Table 2, is this a fair comparison? It seems like there would be size benefits favoring the larger systems especially in terms of infrastructure required no matter the size of the system.

RE: This is a good point! We all agree that the power efficiency is currently an issue with rich uncertainties. More discussions on the uncertainties and future work direction are added in the revision. Please see L225-230. Thanks

SV6: see bottom of the page

Line 307 (257): swlu was italicized but not introduced in section 2.1.2. It appears in capitals as SWLU in Table 1.

RE: The tool name in Table 1 is modified to italicized, consistent with the context now. Thanks.

OK

Line 332 (279): TAMU and QNLM are undefined (QNLM is defined on line 602, TASMU on line 603)

RE: In the revision, TAMU and QNLM are first defined at L116-117. Thanks.

OK

Line 341 (288): It seems that such short runs are not sufficient. Did you make a similar test with more than nine time steps. There remains a high potential for variations to show up

SV6: The discussion you added is not clear to me. Considering also comment by reviewer #2, I think something like the following would be better: "However, considering that weather and climate applications do not maximise the FLOPs provided by the computer, those numbers do not demonstrate that the Sunway TaihuLight system has any real advantage in actual power efficiency. To conclude on this point, more precise calculations taking into account the real FLOPS of the applications and including the human labour cost of the porting and optimisation effort (given that Sunway TaihuLight is a unique system) have to be realised.

SV7: Figure 9 appears in the text after figures 10 and 11. You should rename the figures 10->9, 11->10, 9->11.
SV8: Legend of figure 11: I don't understand the "any two of 4" as you show 6 black-dashed-dotted lines.
SV9: Legend of figure 11: I would remove the "3" in "3 timeseries …"
SV10: Legend of figure 11: The dots of QNLM-TAMU are purple (not pink)

later in the simulation experiments Optimizations were made to achieve 1 SYPD on the Sunway system. Did you attempt to incorporate these changes and optimization techniques back into the original model?

RE: We also compare the results of 1-yr long runs in different perturbation scenarios (as shown in Fig. 10) to comprehend the integrations on Sunway machine. More discussions on the 9 timesteps are added (please see L460-463). To minimize the coding-caused uncertainties in the porting and optimizing process, at this stage, the current work doesn't apply any change to the original code. More statement and discussions are added in the revision. Please see L125-126. Thanks.

OK for the 9 timesteps (modifications are lines 455-457, not 460-463).
Results of 1-yr long runs are shown on Fig. 11, not Fig.10.

Line 372 (316): Can you provide more details or analysis regarding why -O2 fails but -O3 passes? How long were the runs made before comparisons were done?

RE: While more discussions on the metrics of the ECT tool are added in L460-463, the possible reason for which -O2 fails but -O3 passes is given in the revision. Please see L318-319; L460-463.
312-315

OK (modifications are lines 455-457, not 460-463).

**Responses to R2**
This paper provides a comprehensive description of the efforts to port large legacy code of the CESM model to the Sunway TaihuLight processors. The text gives a very detailed description of the improvements made to the model, the parallelization and optimization techniques employed as well as the programming models used (OpenACC and Athread). A pre-industrial simulation over 400 years is being performed and the main result is an optimization from 1 SYPD to 3.4 SYPD. The paper is well structured and provides comprehensive information about the model, and the experiments for reproducibility. The software is open source and referenced within the paper.

General comments: the text is sometimes too dense and hard to follow. (1) I would recommend to interleave the relevant tables and figures within the text, next to the references and the discussion. Otherwise it is hard to follow references to tables and figures that are placed at the end. (2) For a paper running on such a large system (65000CGs), a scalability plot is missing. (3) The paper emphasizes in several places the energy consumption point of view and the advantage of hybrid architectures like TaihuLight. However, there is no real data for this experiment presented, therefore there is no data that support some strong statements presented in the text.

RE: Thanks for the reviewer's thorough examination of our manuscript (MS) and positive comments.

We all agree that your comments are very constructive for us to improve presentation of the MS, and all your major comments and other points have been fully addressed in the revision. Specifically, in the revision, (1) the Tables and Figs. are inserted into the relevant places in the following marked-up version to make it better to read. But due to the format requirement, we did not revise it in the submitted MS; (2) a scalability plot is added; (3) the energy consumption statement is appropriately re-written and more discussions about the uncertainties of the current work on power efficiency are added.

The point-by-point replies are followed.

Line105 (89 in revised MS): the argument is valid only for applications that maximize the FLOPs provided by a computer, which does not hold for weather and climate applications. Same happens with Table 2 which presents general specs measured for different machines. If 3.4 SYPD are achieved with 65000, while the benchmarks with ~11000 Intel processors runs at ~1SYPD I conclude that for the same SYPD, the TaihuLight requires 2 times more processors than the Intel system. How does the energy efficiency of the TaihuLight chip compares to the Intel? I encourage to backup the energy arguments with data from the experiments or make a clearer link. The authors (very rightly) emphasize the large efforts required to port this large model to a new architecture. Here the reviewer is missing a more general discussion about the cost of these effort and performance portability. Was performance portability an important metric of this work? Considering that the TaihuLight is a unique system that didnt go into the market of supercomputing, what is the cost of such refactoring? It would be interesting for the paper to provide a number for financial costs of the porting effort.

RE: We all agree that the power efficiency is currently an issue with rich uncertainties. The statement of power efficiency is modified, and more discussions on the uncertainties and future work direction are added in the revision. Please see L225-230. The discussions on the performance-portability issue are added in the revision. Please see L128-131; L370-374; L375; L387-388; L532-533; L540-543. Thanks a lot!

SV11: for the discussion on efficiency, see SV6 ; for the discussion on performance-portability, please point me to the lines in the Marked-up Manuscript Version where you discuss this issue.

More specific comments:

The introduction is too long. It can be simplified and emphasize the contributions of this work. The related work part of the introduction is well written.

RE: The introduction is condensed, focusing on relevant project now. Please see the new introduction. Thanks.

OK: Introduction went down from ~3 pages to 2.75 pages

line72: I don't think there is any major system in the list of supercomputers with FPGAs.

RE: The related statement is removed. Thanks.

SV12: Can you write down the line that has been removed, I can't find it.

line: 158-159 This is not very correct. The authors already mentioned in the literature NIM that was ported to GPU and XeonPhi. Other models like ICON are GPU ready.

RE: The sentence has been removed in the revision. Thanks.

SV13: Can you write down the line that has been removed, I can't find it.

line 265 (218): This is a very strong statement. It is true that the systems are very different. But from

these differences is not obvious that a GPU system with multiple GPUs connected with a high-throughput low latency NVLink is not more suitable for scientific computations.

RE: The statement has been re-written in a more appropriate way. Please see L218-221.

SV14: The only change I see is "is more plausible" changed to "may be more plausible". Please add something more along the reviewer's remark.

Section 2.2.2: as already mentioned, this refer to general specifications of the system which are for sure different than the FLOP/energy consumption of weather applications. I would consider providing experiment data or removing this section.

RE: Agree that energy consumption is a complex issue and at the current stage, the Sunway machine has not shown any advantage on power efficiency. We may pursue really-greener utilization in the future when more experiences and optimization skills can make the computation with much higher efficiency. Please see L226-230.

See SV6 above

Section 3.1: It is hard to follow the text and what are the major improvement. Considering supporting with data, figures and simplifying the list.

RE: The major improvement of CESM1.3-beta17_sehires38 has been reorganized. Please see the new Section 3.1. Thanks.

OK

**A List of All Relevant Changes**

In this revision, we have addressed all comments proposed by reviewers and made changes at corresponding positions. The revision is further proofread and polished as well. In particular, some of the major changes we have made include,

1) We have added more discussions and analysis referring to several very important issues, such as general performance-portability, power efficiency, speedup and the accuracy of experiments, and so on.
2) We have added more detailed and clear descriptions about the system architecture of the Sunway TaihuLight, including its network architecture, systematic specification, optimization techniques and tools, etc.
3) We have also revised or removed some of the awkward sentences.

Details of every changes we have made are listed in the first part (**Point-by-point Response to the Reviews**).

**A Marked-up Manuscript Version**

---

## Author Response (AR2)

Mark Govett (Referee) mark.w.govett@noaa.gov

General Comments: This was a very well organized and written paper. The paper described efforts to port a large legacy, climate code to the Sunway TaihuLight system. The unique architecture of the Sunway processor was described which helps the reader understand some of the changes that were needed for the climate application to run efficiently. The original code was designed and run on Intel-based processors. Performance and baseline scientific results were made prior to porting the code. The work described detailed efforts to optimize the code, while maintaining sufficient accuracy in the solutions. As the authors admit, determining an acceptable level of scientific accuracy is an ongoing process determined by many factors. With a complex scientific application, thorough testing and evaluation using multiple criteria is needed to build confidence in the solution.

RE: Thank you for the thorough examination on our manuscript (MS) and constructive comments. We agree that all of the comments are very useful for us to improve the presentation of the MS, and we have fully addressed them in the revision.

What follows is a point-by-point reply to each comment. **Please refer to marked-up copy for line numbers.**

Specific Comments:
(1) The authors describe extensive efforts to optimize the code, which included many common techniques. You spent a lot of time optimizing for the Sunway processor, but did not apply that level of effort to the original code. Some of the changes could have been applied to original code, making the comparison of performance more fairly represented.

RE: This is a good point that needs to be addressed in the MS. The work presented in this paper represents the first step of running CESM on the new architecture machine. To minimize the coding uncertainties, we keep the original CPU code unchanged for both accuracy verification and performance evaluation. It is true that many of the applied techniques can be generalized to CPU or even GPU architectures. That would be the focus of a later-stage study, which is to extract both general practices and tools that would help the transition from current multi-core CPUs to many-core accelerators. Redesigning some of the original algorithms is expected to further improve the efficiency. Some clarifications

and discussions are added in the revision. Please see L43-44; L125-126; L494-496; L635-638. Thanks a lot!

(2) The work appears specifically designed to target a single system with a unique processor. Was performance portability considered as a factor in this effort? Could the modified code run on an Intel-based system and how did the results compare to the original code. Given the fine-grain nature of the parallelization, would GPUs or high core count CPU processors be a target for this work? Addressing performance-portability would make the impact of this work much greater than the results you achieved targeting a single system.

RE: We agree that addressing the issue of general performance-portability is an important and interesting topic in using heterogeneous many-core HPC systems and should be further clarified and discussed more in the MS. In the revision we expanded the discussion of this topic. Please see L128-131; L418-421; L426; L438-439; L627-628; L635-638. Thanks a lot!

(3) The performance impact of the different types of code optimizations you made were not described. This would be a useful way to determine the tradeoff between portability and performance. For example, specific optimizations described in the Stage 1-3 optimizations were closely aligned to the Sunway processor. How much performance benefit were there for each of the stages and was it applied to a large portion of the code?

RE: Combined with comment (2), this is a very good comment addressing the general performance-portability issue. The description of performance impact of the different types of code optimizations is now added. Please see L418-421; L426; L438-439. The three-stage optimizations described in section are proposed to solve a specific prefix summation calculation using register communication. Our method is 27 times faster than the original code that has only a single loop. More discussions about its performance portability are added in the revision. Please see L418-421.

(4) Regarding porting the code to the TaihuLight system, your approach seemed to be first port the code to two Intel based systems (TAMU and QNLM). It was unclear why you felt the need to port to both systems. Further, your comparisons were made after only 9 timesteps seemed arbitrary and perhaps no sufficient. Please include some justification. It was also unclear what fields were compared in the UF-CAM-ECT test. A summary of the relevant details from the paper would be useful here.

RE: Given the totally new architecture of the Sunway machine, to minimize the uncertainties of code porting, before we port the CESM-HR to the Sunway TaihuLight, we first ran and tested its correctness on the Intel multi-core supercomputing platforms available to iHESP, serving as the first benchmark. The justification and adjusted statement are made in the revision. Please see L314-318.

More descriptions and discussions about UF-CAM-ECT tools are added in the revision. Please see L324-326; L533-536. Thanks.

(5) Are there references for the tools given in Table 1? Most of the tools listed were not referenced in the manuscript. They should be either introduced to the reader in the paper if they add value to the manuscript. For example, you could state how you used them and how it helped identify

RE: The point is well taken! One more column describing the role of each tool in this project is now added in the Table 1. More reference information about Table 1 is added in the revision. Please see the new Table 1 and L199-200. Thanks.

Technical Corrections:
Line 103-105: Awkward sentence. Perhaps break into two sentences?

RE: The entire sentence disappears because of concise introduction in the revision. Thanks.

Line140 (104 in revised MS): Old reference (Govett, 2010) should be replaced with a more comprehensive paper (Govett, et al. 2015) in the Bulletin of the AMS: https://doi.org/10.1175/BAMSD-15-00278.1

RE: Done. Please see L104; L711-713. Thanks.

Lines 212-218 (176-181): The authors don't describe what the speed and bandwidth of connections between super-nodes, within a cabinet, and between cabinets. This is essential in understanding the limitations of the Sunway TaihuLight system at scale.

RE: Good comment! The TaihuLight compute nodes are connected via a 2-level InfiniBand network. A single-switch with full bisection bandwidth connects all 256 nodes within a super-node, while a fat-tree with 1/4 of the full bisection bandwidth connects all super-nodes (as shown the following Figure). Table II shows measurements of bisection communication bandwidth at different levels of the system.

[Figure]

**TABLE II**
**TAIHULIGHT SPECIFICATIONS AND BANDWIDTH MEASURED**

| Component | Configurations | BW measured (% peak) |
|---|---|---|
| MPE | 1.45 GHz, 32/256KB L1/L2 | DRAM 8.0 GB/s (80%) |
| CPE | 1.45 GHz, 64KB SPM | Reg. 630 GB/s (85%) |
| CG | 1 MPE + 64 CPEs | DRAM 28.98 GB/s (85%) |
| Node | 1 CPU (4 CGs), 4×8GB RAM | Net: 6.04 GB/s (89%) |
| Super Node | 256 nodes, FDR 56 Gbps IB | Bisection 1.5 TB/s (72%) |
| TaihuLight | 160 supernodes | Bisection 49.2 TB/s (68.6%) |
| Agg. memory | 1.3 PB | 4.6 PB/s (85%) |
| Storage | 5.2 PB (Online2) | 70 GB/s (54.35%) |

Such essential information is added in the revision. Please see L181-184. Thanks.

Line 243 (204): change "details Section 3.3" to: "detail in Section 3.3" Line 245-250 (208-213): unclear if CPE based parallelism is with MPI or something else??? Line 251 (215): change "details" to "detail"

RE: Line 204: Done; Line 215: Done.
Line 208-213: The statements are revised for clarification. Please see L212-215.

Lines 260-266 (228-231): It appears that an MPI-based intelligent programming model would work here. The intelligence would be knowing when comms within a CPU task group or to an MPE are needed.

RE: Thanks for reminding the clarification. The statement is further clarified based on the comment. Please see L231-234.

Lines 268-272 (236-239): Regarding the power efficiency comparison in Table 2, is this a fair comparison? It seems like there would be size benefits favoring the larger systems especially in terms of infrastructure required no matter the size of the system.

RE: This is a good point! We all agree that the power efficiency is currently an issue with rich uncertainties. More discussions on the uncertainties and future work direction are added in the revision. Please see L239-244. Thanks

Line 307 (271): swlu was italicized but not introduced in section 2.1.2. It appears in capitals as SWLU in Table 1.

RE: The tool name in Table 1 is modified to italicized, consistent with the context now. Thanks.

Line 332 (308): TAMU and QNLM are undefined (QNLM is defined on line 602, TASMU on line 603)

RE: In the revision, TAMU and QNLM are first defined at L116-117. Thanks.

Line 341 (316): It seems that such short runs are not sufficient. Did you make a similar test with more than nine time steps. There remains a high potential for variations to show up

later in the simulation experiments Optimizations were made to achieve 1 SYPD on the Sunway system. Did you attempt to incorporate these changes and optimization techniques back into the original model?

RE: We also compare the results of 1-yr long runs in different perturbation scenarios (as shown in Fig. 11) to comprehend the integrations on Sunway machine. More discussions on the 9 timesteps are added (please see L533-536). To minimize the coding-caused uncertainties in the porting and optimizing process, at this stage, the current work doesn't apply any change to the original code. More statement and discussions are added in the revision. Please see L125-126. Thanks.

Line 372 (345): Can you provide more details or analysis regarding why -O2 fails but -O3 passes? How long were the runs made before comparisons were done?

RE: While more discussions on the metrics of the ECT tool are added in L533-536, the possible reason for which -O2 fails but -O3 passes is given in the revision. Please see L352-353; L533-536.

**Responses to R2**
This paper provides a comprehensive description of the efforts to port large legacy code of the CESM model to the Sunway TaihuLight processors. The text gives a very detailed description of the improvements made to the model, the parallelization and optimization techniques employed as well as the programming models used (OpenACC and Athread). A pre-industrial simulation over 400 years is being performed and the main result is an optimization from 1 SYPD to 3.4 SYPD. The paper is well structured and provides comprehensive information about the model, and the experiments for reproducibility. The software is open source and referenced within the paper.

General comments: the text is sometimes too dense and hard to follow. (1) I would recommend to interleave the relevant tables and figures within the text, next to the references and the discussion. Otherwise it is hard to follow references to tables and figures that are placed at the end. (2) For a paper running on such a large system (65000CGs), a scalability plot is missing. (3) The paper emphasizes in several places the energy consumption point of view and the advantage of hybrid architectures like TaihuLight. However, there is no real data for this experiment presented, therefore there is no data that support some strong statements presented in the text.

RE: Thanks for the reviewer's thorough examination of our manuscript (MS) and positive comments.

We all agree that your comments are very constructive for us to improve presentation of the MS, and all your major comments and other points have been fully addressed in the revision. Specifically, in the revision, (1) the Tables and Figs. are inserted into the relevant places in the following marked-up version to make it better to read. But due to the format requirement, we did not revise it in the submitted MS; (2) a scalability plot is added; (3) the energy consumption statement is appropriately re-written and more discussions about the uncertainties of the current work on power efficiency are added.

The point-by-point replies are followed. **Please refer to marked-up copy for line numbers.**

Line105 (89 in revised MS): the argument is valid only for applications that maximize the FLOPs provided by a computer, which does not hold for weather and climate applications. Same happens with Table 2 which presents general specs measured for different machines. If 3.4 SYPD are achieved with 65000, while the benchmarks with ~11000 Intel processors runs at ~1SYPD I conclude that for the same SYPD, the TaihuLight requires 2 times more processors than the Intel system. How does the energy efficiency of the TaihuLight chip compares to the Intel? I encourage to backup the energy arguments with data from the experiments or make a clearer link. The authors (very rightly) emphasize the large efforts required to port this large model to a new architecture. Here the reviewer is missing a more general discussion about the cost of these effort and performance portability. Was performance portability an important metric of this work? Considering that the TaihuLight is a unique system that didnt go into the market of supercomputing, what is the cost of such refactoring? It would be interesting for the paper to provide a number for financial costs of the porting effort.
RE: We all agree that the power efficiency is currently an issue with rich uncertainties. The statement of power efficiency is modified, and more discussions on the uncertainties and future work direction are added in the revision. Please see L238-244. The discussions on the performance-portability issue are added in the revision. Please see L128-131; L418-421; L426; L438-439; L627-628; L633-638. Thanks a lot!

More specific comments:

The introduction is too long. It can be simplified and emphasize the contributions of this work. The related work part of the introduction is well written.
RE: The introduction is condensed, focusing on relevant project now. Please see the new introduction. Thanks.

line72: I don't think there is any major system in the list of supercomputers with FPGAs.
RE: The related statement is removed. Thanks.

line: 158-159 This is not very correct. The authors already mentioned in the literature NIM that was ported to GPU and XeonPhi. Other models like ICON are GPU ready.
RE: The sentence has been removed in the revision. Thanks.

line 265 (231): This is a very strong statement. It is true that the systems are very different. But from

these differences is not obvious that a GPU system with multiple GPUs connected with a high-throughput low latency NVLink is not more suitable for scientific computations.

RE: The statement has been re-written in a more appropriate way. Please see L231-234.

Section 2.2.2: as already mentioned, this refer to general specifications of the system which are for sure different than the FLOP/energy consumption of weather applications. I would consider providing experiment data or removing this section.

RE: Agree that energy consumption is a complex issue and at the current stage, the Sunway machine has not shown any advantage on power efficiency. We may pursue really-greener utilization in the future when more experiences and optimization skills can make the computation with much higher efficiency. Please see L239-244.

Section 3.1: It is hard to follow the text and what are the major improvement. Considering supporting with data, figures and simplifying the list.

RE: The major improvement of CESM1.3-beta17_sehires38 has been reorganized. Please see the new Section 3.1. Thanks.

**A List of All Relevant Changes**

In this revision, we have addressed all comments proposed by reviewers and made changes at corresponding positions. The revision is further proofread and polished as well. In particular, some of the major changes we have made include,

1) We have added more discussions and analysis referring to several very important issues, such as general performance-portability, power efficiency, speedup and the accuracy of experiments, and so on.
2) We have added more detailed and clear descriptions about the system architecture of the Sunway TaihuLight, including its network architecture, systematic specification, optimization techniques and tools, etc.
3) We have also revised or removed some of the awkward sentences.

Details of every changes we have made are listed in the first part (**Point-by-point Response to the Reviews**).

**A Marked-up Manuscript Version**

[revised manuscript text omitted]

¶
¶
-------------------------------------------------Page Break-----------------------------------------------------
¶                                                                              ... [1]

Page 40: [1] Deleted               Gan Lin               6/12/20 2:09:00 PM

---

## Author Response (AR3)

**Point-by-point Response to the Reviews**

However your manuscript is still missing the following sections: Code availability, Author Contribution, Competing Interests (these sections appeared in version 2 of your manuscript but have disappeared since then). Your Code availability section has to cover the CESM code and all tools; in that sense, the code availability section of version 2 of your manuscript is not complete. You may have to add a Data Availability section too.

Code availability, Author Contribution, Competing Interests, and Data Availability sections are added in the new submission.

**A List of All Relevant Changes**

In this revision, we have added the following parts according to the requests of the editor. Section 6 Code availability, Section 7 Data availability, Section 8 Author contributions, Section 9 Competing interests.

**A Marked-up Manuscript Version**

[revised manuscript text omitted]